

# Boreal forest BVOCs exchange: emissions versus in-canopy sinks

Putian Zhou[1], Laurens Ganzeveld[2], Ditte Taipale[3,4], Üllar Rannik[1], Pekka Rantala[1], Matti P. Rissanen[1], Dean Chen[1], and Michael Boy[1]

[1]University of Helsinki, Department of Physics, P.O. Box 64, FI-00014, University of Helsinki, Finland
[2]Meteorology and Air Quality (MAQ), Department of Environmental Sciences, Wageningen University and Research Centre, Wageningen, Netherlands
[3]University of Helsinki, Department of Forest Sciences, P.O. Box 27, FI-00014, University of Helsinki, Finland
[4]Estonian University of Life Sciences, Department of Plant Physiology, Kreutzwaldi 1, EE-51014, Estonia

*Correspondence to:* Putian Zhou (putian.zhou@helsinki.fi)

**Abstract.** A multi-layer gas dry deposition model has been developed and implemented into a 1-dimensional chemical transport model SOSAA (a model to Simulate the concentrations of Organic vapours, Sulphuric Acid and Aerosols) to calculate the dry deposition velocities for all the gas species included in the chemistry scheme. The new model was used to analyse in-canopy sources and sinks, including gas emissions, chemical production and loss, dry deposition and turbulent transport of

5 12 featured biogenic volatile organic compounds (BVOCs) or groups of BVOCs (e.g., monoterpenes, isoprene+2-methyl-3-buten-2-ol (MBO), sesquiterpenes and oxidation products of mono- and sesquiterpenes) in July, 2010 at the boreal forest site SMEAR II (Station to Measure Ecosystem-Atmosphere Relations II). According to the significance of modeled monthly-averaged individual source and sink terms inside the canopy, the selected BVOCs were classified into five categories: (1) most of emitted gases are transported out of the canopy (monoterpenes, isoprene+MBO), (2) chemical reactions remove a significant

portion of emitted gases (sesquiterpenes), (3) bidirectional fluxes occur since both emission and dry deposition are crucial for the in-canopy concentration tendency (acetaldehyde, methanol, acetone, formaldehyde), (4) gases removed by deposition inside the canopy are compensated by the gases transported from above the canopy (acetol, pinic acid, $\beta$-caryophyllene's oxidation product BCSOZOH), and finally (5) the chemical production is comparable to the sink by deposition (isoprene's oxidation products ISOP34OOH and ISOP34NO3).

Most of the simulated sources and sinks were located above about 4 m for oxidation products and above about 8 m for emitted species except formaldehyde. In addition, soil deposition (including deposition onto understory vegetation) contributed 11 – 61% to the overall in-canopy deposition. The emission sources peaked at about 14 – 16 m which was higher than 10 m where the maximum of dry deposition onto overstorey vegetation was located.

This study provided a method to enable the quantification of the exchange between atmosphere and biosphere for numerous

BVOCs, which could be applied in large-scale models in future. With this more explicit canopy exchange modeling system this study analysed both the temporal and spatial variations of individual in-caonpy sources and sinks, as well as their combined effects on driving BVOCs exchange. Twelve featured BVOCs or BVOC groups were analyzed in this study, more compounds could also be investigated similarly by being classified into the five categories.



# 1 Introduction

Boreal forests emit a large amount of biogenic volatile organic compounds (BVOCs), which include monoterpenes ($C_{10}H_{16}$), isoprene ($C_5H_8$), sesquiterpenes ($C_{15}H_{24}$), methanol ($CH_3OH$), acetone ($CH_3C(O)CH_3$), acetaldehyde ($CH_3CHO$) and many others (Rinne et al., 2009; Guenther et al., 2006, 2012). Once emitted, BVOCs in the atmosphere can be transformed by

reacting with the hydroxyl radical (OH), ozone ($O_3$) or nitrate radical ($NO_3$), producing oxygenated volatile organic compounds (OVOCs). The OVOCs can be oxidized to carbon monoxide (CO) or carbon dioxide ($CO_2$) by further chemical reactions, participate in secondary organic aerosol (SOA) formation, or deposit onto surfaces (Goldstein and Galbally, 2007).

BVOC emissions from boreal pine forests have been investigated extensively in previous studies (e.g., Taipale et al., 2011; Bäck et al., 2012; Aalto et al., 2014). For example, the seasonal branch-scale measurements of emissions of monoterpenes

and sesquiterpenes from Scots pine trees were reported in Tarvainen et al. (2005) and Hakola et al. (2006). More compounds besides monoterpenes, including methanol, acetaldehyde and acetone, were measured by Rinne et al. (2007) at an ecosystem scale. Recently, Rantala et al. (2015) revised the fluxes of isoprene, monoterpenes, and several OVOCs over a boreal forest canopy based on long-term measurements from 2010 to 2013. In addition, the boreal forest floor also plays a significant role in BVOC exchange (e.g., Mäki et al., 2017). Aaltonen et al. (2011) reported the emissions of monoterpenes (5 $\mu$g m$^{-2}$ h$^{-1}$),

isoprene (0.050 $\mu$g m$^{-2}$ h$^{-1}$) and sesquiterpenes (0.045 $\mu$g m$^{-2}$ h$^{-1}$) from ground vegetation and soil. Compared to the ecosystem-scale fluxes, the fluxes of BVOCs (e.g., methanol acetaldehyde, acetone and monoterpenes) from forest floor were about one to two orders of magnitude lower (Aaltonen et al., 2013).

The BVOCs emitted or produced by chemical reactions are dispersed by turbulent air motion, being possibly uptaken by vegetation surfaces which is controlled by different dry deposition pathways, or partly transported into higher atmosphere.

The majority of the BVOCs diffuse between the ambient air and intercellular air space via stomata with the direction of this exchange being dependent on the concentration gradient. For those BVOCs being metabolized rapidly, enzymatically or non-enzymatically, in the intercellular air space one would anticipate to have stomatal deposition with the uptake rate being determined by stomatal conductance. For those BVOCs with a less efficient intercellular air space destruction or actually a production mechanism, the actual direction and efficiency of the stomatal exchange depends on the leaf-sale compensation

point. A small part of them, especially the hydrophobic ones, can be absorbed directly by the cuticle or diffusing into the leaf through the cuticle (Niinemets et al., 2014). However, in contrast to our understanding of BVOC emissions, understanding the role, magnitude as well as mechanisms of dry deposition of BVOCs is still quite poor due to limited measurement techniques, although it may play a significant role in estimating BVOC fluxes (Goldstein and Galbally, 2007; Hallquist et al., 2009). Recently, more studies have focused on this topic. Karl et al. (2010) observed substantial dry deposition removal of several

OVOCs in field measurements. Park et al. (2013) directly observed bidirectional fluxes for 494 organic ions and daily mean net deposition for 186 organic ions over an orange grove and recently Nguyen et al. (2015) observed dominant daytime dry deposition fluxes for small, saturated OVOCs derived from isoprene and monoterpene oxidation during summer. Furthermore, the significance of dry deposition of OVOCs has been revealed by various model systems. For example, a regional simulation over US during summer 2010 indicated that removal of 60-75% of the tropospheric SOA burden was achieved by dry and wet



deposition of condensible organic vapors which was more important than the direct SOA scavenging (Hodzic et al., 2014). Knote et al. (2015) also found that the modeled SOA concentrations over the continental US would be about 50% larger than the observation when not considering dry and wet deposition of semi-volatile organic compounds (SVOCs).

Inside the canopy, the emissions, deposition and the chemical reactions all-together result in net upward or downward fluxes of different BVOCs. Compared to measurements, numerical models appear the only feasible way to assess individual source and sink terms within the canopy. Several gas dry deposition models have been developed since the widely used deposition model proposed by Wesely (1989). However, only few models focused on dry deposition processes of BVOCs until recently not only possibly because of a lack of recognition of deposition being a potentially important BVOC sink but also because of limited experimental information to constrain the dry deposition calculations. One of the difficulties is in obtaining Henry's law constants or effective Henry's law constants for various organic compounds. The models GECKO-A (Generator of Explicit Chemistry and Kinetics of Organics in the Atmosphere) and GROMHE (Raventos-Duran et al., 2010, GROup contribution Method for Henry's law Estimate;) were used to calculate the effective Henry's law constants for organic compounds in Hodzic et al. (2014) and Knote et al. (2015). Nguyen et al. (2015) used the Henry's law constants with modifications of the original dry deposition model from Wesely (1989). All of the models used in these studies by Hodzic et al. (2014), Knote et al. (2015) and Nguyen et al. (2015) applied the big-leaf approach (Hicks et al., 1987), which assumed the whole canopy as one single big leaf and thus did not reveal more details on the actual mechanisms inside the canopy that ultimately determine the effective BVOC exchange fluxes. The deposition process was also included in several multi-layer canopy exchange models for field site studies, e.g. CACHE (Canopy Atmospheric CHemistry Emission model; Bryan et al., 2012), FORCAsT (FORest Canopy Atmosphere Transfer) 1.0 (Ashworth et al., 2015), CAFE (The Chemistry of Atmosphere-Forest Exchange model) (Wolfe and Thornton, 2011; Wolfe et al., 2011) and a multi-layer exchange model used for global-scale canopy process studies (Ganzeveld et al., 2002, 2010). However, in these studies quite a large number of BVOCs for which deposition can potentially be a relevant sink, have been excluded. Moreover, detailed deposition contributions for BVOCs have not been analysed. Both of them motivated this study.

A multi-layer gas dry deposition model has been developed in this study based on several models in previous studies (Wesely, 1989; Ganzeveld et al., 2002; Nguyen et al., 2015; Zhou et al., 2017). It has been implemented into SOSAA (a model to Simulate the concentrations of Organic vapours, Sulphuric Acid and Aerosols; Boy et al., 2011; Zhou et al., 2014) to evaluate emissions, chemistry, dry deposition and turbulent transport processes for BVOCs within the canopy. The model was applied to study boreal forest BVOC exchange and particularly BVOC deposition, for the time period of July, 2010 at SMEAR II (Station to Measure Ecosystem-Atmosphere Relations II) also having access to a large number of emission and other complementory measurements collected during an intensive field campaign (HUMPPA, Williams et al., 2011) in July at this site. In this study, we aim to provide a general multi-layer parametrization model to calculate the dry deposition velocities of large amounts of gas species included in a detailed chemistry scheme. By using this newly implemented model and conducting an extensive evaluation by comparison with the 2010 field observations, we have investigated individual in-canopy sources and sinks of several featured BVOCs at a boreal forest, and thus give a new insight of how different processes inside the canopy contribute to the BVOC exchange between atmosphere and biosphere.



## 2 Measurements

### 2.1 Measurement site

All the observation data were measured at SMEAR II in Hyytiälä, Finland (61°51'N, 24°17'E, 181 m a.m.s.l., UTC+02). The site was situated in a relatively homogeneous boreal forest, mainly composed of Scots pine (*Pinus sylvestris*), but with some

Norway spruce (*Picea abies*) and broadleaved trees (Bäck et al., 2012). The canopy height ($h_c$) was about 18 m in 2010. The all-sided leaf area index (LAI) of the canopy was about 7.5 $m^2 \, m^{-2}$ with ~6.0 $m^2 \, m^{-2}$ overstory vegetation mainly consisting of tree leaves, ~0.5 $m^2 \, m^{-2}$ understory vegetation consisting of lingonberry (*Vaccinium vitis-idaea*) and blueberry (*Vaccinium myrtillus*), as well as ~1 $m^2 \, m^{-2}$ of cover by mosses on the ground (Launiainen et al., 2013). More detailed description of this site has been reported in Hari and Kulmala (2005), Haapanala et al. (2007) and Ilvesniemi et al. (2009).

### 2.2 Measurement method

### 2.2.1 Meteorological data

In this study, the measured meteorological data are either used as model input to constrain the simulations in a realistic range or used for analysis. The air temperature ($T$) was measured by Pt100 sensor at 4.2 m, 8.4 m, 16.8 m, 33.6 m, 50.4 m and 67.2 m above the ground level. The water vapor mixing ratio was measured by Li-Cor LI-840 infrared light absorption analyser at the

same height levels. The relative humidity (RH) was then calculated from the water vapor mixing ratio and the air temperature. The photosynthetically active radiation (PAR, 400-700 nm) was measured at two heights with different instruments, one was measured by Li-Cor Li-190SZ quantum sensor at 18 m and the other was measured by the array of four Li-Cor Li-190SZ sensors at 0.6 m. The Reeman MB-1 net radiometer was installed to measure the net radiation ($R_{net}$) at 67 m. The sensible and latent heat fluxes ($SH$ and $LE$) were measured at 23 m by Gill Solent 1012R and Li-Cor LI-6262 gas analyzer, and the

soil heat flux ($G_{soil}$) was measured by Hukseflux HFP01 heat flux sensors at the ground surface.

### 2.2.2 VOC measurement

The concentrations of 27 different masses (mass-to-charge ratio, m/z) of BVOCs were measured by the proton transfer reaction quadrupole mass spectrometer (PTR-MS, manufactured by Ionicon Analytik GmbH, Innsbruck, Austria) at the same six height levels as the air temperature (Rantala et al., 2015). The fluxes of BVOCs, based on 45-minute averages, were computed with

the surface-layer-profile method (Rannik, 1998; Rantala et al., 2014). Then the fluxes were filtered as suggested in Rantala et al. (2015), according to which the data points were removed from the time series when $\zeta$ (Obukhov stability parameter) < -2, $\zeta$ > 1 or $u_*$ (friction velocity at 23 m) < 0.2 m s$^{-1}$. Since only one-month data were used for comparison with the model results, we did not disregard the outliers and did not apply the gap-filling. Finally, for each compound one data point was filtered out from 164 measurement data points.

Out of 27 measured masses, 7 single or group compounds were identified and used in this study, which were monoterpenes (m/z 137), isoprene (m/z 69), 2-methyl-3-buten-2-ol (MBO, $C_5H_{10}O$, m/z 87), methanol (m/z 33), acetaldehyde (m/z 45),



acetone (m/z 59) and formaldehyde (HCHO, m/z 31). MBO fragmented heavily on m/z 69, thus its concentrations were not calibrated. Therefore, the m/z 69 was not only related to isoprene but also to the fragments of MBO (Rantala et al., 2015). Hence in this study isoprene and MBO are analysed together as one group, written as isoprene+MBO. It should also be noted that there is a large uncertainty in the formaldehyde flux measurements according to Rantala et al. (2015). This is mostly due

to the high sensitivity of formaldehyde to water vapor, as their proton affinities are near, but also because the concentrations of formaldehyde were not calibrated.

## 3 Model description

### 3.1 SOSAA

SOSAA is a one-dimensional (1-D) column model which was first developed by Boy et al. (2011) and applied in several subse-

10 quent studies since then (e.g., Kurtén et al., 2011; Mogensen et al., 2011; Bäck et al., 2012; Boy et al., 2013; Smolander et al., 2014; Zhou et al., 2015; Mogensen et al., 2015; Zhou et al., 2017). SOSAA is written in Fortran90 and able to run in parallel in superclusters. The current version has coupled five modules. The meteorology module is derived from SCADIS (SCAlar DIStribution; Sogachev et al., 2002) which is originally a 3-dimensional (3-D) boundary layer meteorology model. The BVOC emissions from the forest ecosystem are computed by MEGAN (Model of Emissions of Gases and Aerosols from Nature;

Guenther et al., 2006). The chemistry module codes are created by KPP (Kinetic PreProcessor; Damian et al., 2002) based on the chemical mechanisms generated by MCMv3.2 (Master Chemical Mechanism version 3.2; http: //mcm.leeds.ac.uk/MCM) (Jenkin et al., 1997; Saunders et al., 2003; Jenkin et al., 2012). The MCM names (if available) of all the species mentioned in this study are listed in Table 1, which also shows the abbreviation names used in this study (context names), the chemical names and formulas. The aerosol module is based on UHMA (University of Helsinki Multicomponent Aerosol model; Korho-

nen et al., 2004), which describes the nucleation, condensation, coagulation and deposition of aerosol particles. The gaseous dry deposition module was first introduced in Zhou et al. (2017) mostly focusing on $O_3$ dry deposition. In this study it is extended for all modeled gaseous compounds.

### 3.2 Gas dry deposition model

#### 3.2.1 Basic equations

The gas dry deposition model is based on the $O_3$ dry deposition model described in Zhou et al. (2017). For each model layer, the deposition flux ($F$) of gas $X$ is calculated as

$$F = -[X](\text{LAD}\Delta z V_{dveg} + A_s \Delta z V_{dsoil}) \tag{1}$$

$$V_{dveg} = V_{dveg}(r_b, r_{stm}, r_{mes}, r_{cut}, r_{ws}, f_{wet}) \tag{2}$$

$$V_{dsoil} = V_{dsoil}(r_{bs}, r_{soil}) \tag{3}$$



Table 1: A list of the chemical compounds or groups mentioned in this study, with their context names (abbreviation names used in the text), chemical names, MCM names, formulas and remarks.

| Context name | Chemical name | MCM name | Formula | Remark |
|---|---|---|---|---|
| O($^1$D) | excited state atomic oxygen | O1D | O($^1$D) | |
| O($^3$P) | ground state atomic oxygen | O | O($^3$P) | |
| SO$_2$ | sulfur dioxide | SO2 | SO$_2$ | |
| O$_3$ | ozone | O3 | O$_3$ | |
| NO$_2$ | nitrogen dioxide | NO2 | NO$_2$ | |
| NO | nitric oxide | NO | NO | |
| NH$_3$ | ammonia | NH3 | NH$_3$ | |
| HONO | nitrous acid | HONO | HONO | |
| HNO$_3$ | nitric acid | HNO3 | HNO$_3$ | |
| OH | hydroxyl radical | OH | HO | |
| HO$_2$ | hydroperoxyl radical | HO2 | HO$_2$ | |
| H$_2$O$_2$ | hydrogen peroxide | H2O2 | H$_2$O$_2$ | |
| PAN | peroxyacetyl nitrate | PAN | CH$_3$C(O)OONO$_2$ | |
| peracetic acid | peracetic acid | CH3CO3H | CH$_3$CO$_3$H | |
| glyoxal | glyoxal | GLYOX | OHCCHO | |
| methylglyoxal | methylglyoxal | MGLYOX | CH$_3$C(O)CHO | |
| glycolaldehyde | glycolaldehyde | HOCH2CHO | HOCH$_2$CHO | |
| 2-hydroxy-3-methylbut-3-enal | 2-hydroxy-3-methylbut-3-enal | HC4CHO | CH$_3$C(CH$_2$)CH(CHO)OH | |
| MVK | methyl vinyl ketone | MVK | CH$_3$C(O)CH=CH$_2$ | |
| MACR | methacrolein | MACR | CH$_3$C(CH$_2$)CHO | |
| ROOH | N/A | N/A | N/A | organic hydrogen peroxides |
| isoprene | isoprene | C5H8 | C$_5$H$_8$ | |
| monoterpenes | monoterpenes | N/A | C$_{10}$H$_{16}$ | a class of terpenes, including $\alpha$-pinene, $\Delta^3$-carene, $\beta$-pinene, etc. |
| $\alpha$-pinene | $\alpha$-pinene | APINENE | C$_{10}$H$_{16}$ | |
| $\beta$-pinene | $\beta$-pinene | BPINENE | C$_{10}$H$_{16}$ | |
| $\Delta^3$-pinene | $\Delta^3$-pinene | N/A | C$_{10}$H$_{16}$ | |
| myrcene | myrcene | N/A | C$_{10}$H$_{16}$ | |
| sabinene | sabinene | N/A | C$_{10}$H$_{16}$ | |
| ocimene | ocimene | N/A | C$_{10}$H$_{16}$ | |
| limonene | limonene | LIMONENE | C$_{10}$H$_{16}$ | |
| 1,8-cineole | 1,8-cineole | N/A | C$_{10}$H$_{18}$O | |
| OMT | N/A | N/A | other minor monoterpenes | |
| sesquiterpenes | sesquiterpenes | N/A | C$_{15}$H$_{24}$ | a class of terpenes, including $\beta$-caryophyllene, farnesene, etc. |
| $\beta$-caryophyllene | $\beta$-caryophyllene | BCARY | C$_{15}$H$_{24}$ | |





| Context name | Chemical name | MCM name | Formula | Remark |
|---|---|---|---|---|
| farnesene | farnesene | N/A | $C_{15}H_{24}$ | |
| OSQ | N/A | N/A | other minor sesquiterpenes | |
| MBO | 2-methyl-3-buten-2-ol | MBO | $C_5H_{10}O$ | |
| methanol | methanol | CH3OH | $CH_3OH$ | |
| ethanol | ethanol | C2H5OH | $CH_3CH_2OH$ | |
| formaldehyde | formaldehyde | HCHO | HCHO | |
| acetaldehyde | acetaldehyde | CH3CHO | $CH_3CHO$ | |
| acetone | acetone | CH3COCH3 | $CH_3COCH_3$ | |
| acetol | acetol, hydroxyacetone | ACETOL | $CH_2OHC(O)CH_3$ | |
| pinic acid | pinic acid | PINIC | $C_9H_{14}O_4$ | oxidation product of $\alpha$-pinene |
| BCSOZOH | N/A | BCSOZOH | $C_{15}H_{26}O_5$ | oxidation product of $\beta$-caryophyllene |
| ISOP34NO3 | N/A | ISOP34NO3 | $C_5H_9ONO_3$ | oxidation product of isoprene |
| ISOP34OOH | N/A | ISOP34OOH | $C_5H_{10}O_3$ | oxidation product of isoprene |

where $[X]$ is the concentration of gas species $X$, $\Delta z$ is the layer thickness. LAD is the all-sided leaf area density at layer $i$. $A_s$ represents the soil area index (Eq. 17 in Zhou et al., 2017). $V_{dveg}$ is the vegetation layer-specific conductance which is a function of $r_b$ (quasi-laminar boundary layer resistance), $r_{stm}$ (stomatal resistance), $r_{mes}$ (mesophyllic resistance), $r_{cut}$ (dry cuticular resistance), $r_{ws}$ (resistance to leaf wet skin) and $f_{wet}$ (fraction of wet skin on leaf surface) (see Eqs. 8, 10 – 13 in Zhou et al. (2017)). $V_{dsoil}$ is the soil conductance which is a function of $r_{bs}$ (soil boundary layer resistance) and $r_{soil}$ (soil resistance) (see Eq. 9 in Zhou et al. (2017)).

$r_b$ and $r_{bs}$ are related to both the micro-meteorological quantities and gas properties. For gas $X$, $r_b$ is computed assuming forced convection in the quasi-laminar boundary layer above leaf surface (Grace et al., 1980; Meyers, 1987),

$$r_b = \frac{\mathrm{Sc}^{2/3}}{0.66\nu^{1/2}} \sqrt{\frac{l_d}{U}} \tag{4}$$

$$\mathrm{Sc} = \frac{\nu}{D_X} \tag{5}$$

$$D_X = D_{\mathrm{H2O}} \sqrt{\frac{M_{\mathrm{H2O}}}{M_X}} \tag{6}$$

where Sc is the Schmidt number for gas $X$ defined as the ratio of kinematic viscosity for air ($\nu = 1.59 \times 10^{-5}$ m$^2$ s$^{-1}$) and molecular diffusivity ($D_X$). $D_X$ is then estimated with respect to $D_{\mathrm{H2O}}$ ($2.4 \times 10^{-5}$ m$^2$ s$^{-1}$) according to Graham's law using the molar mass ratio between water vapor ($M_{\mathrm{H2O}}$) and $X$ ($M_X$). $l_d$ (0.07 m) is the characteristic length scale of a leaf along the free-stream wind. $U$ is the horizontal wind speed above the sublayer of leaf surface. $r_{bs}$ is calculated as (Nemitz et al., 2000; Launiainen et al., 2013).

$$r_{bs} = \frac{\mathrm{Sc} - \ln(\delta_0/z_*)}{\kappa u_{*g}} \tag{7}$$

$$\delta_0 = \frac{D_X}{\kappa u_{*g}} \tag{8}$$





where $\delta_0$ is the height above ground where turbulent eddy diffusivity and molecular diffusivity are equal to each other. $z_*$ (0.1 m) is the height up to which the logarithmic wind profile is assumed. $\kappa$ (0.41) is the von Kármán constant and $u_{*g}$ is the friction velocity at the ground surface.

In order to obtain other resistances to vegetation and soil surfaces for all the compounds in the chemistry scheme, a modified parameterization method derived from Wesely (1989) and Nguyen et al. (2015) is applied. Hence,

$$r_{stm} = \frac{D_{H2O}}{D_X} r_{stm,H2O} \tag{9}$$

$$r_{mes} = \left( \frac{H}{50RT_l} + 100f_0 \right)^{-1} \tag{10}$$

$$r_{cut} = \left( \frac{10^{-4}H}{RT_l} + f_0 \right)^{-1} r_{cut,O3} \tag{11}$$

$$r_{ws} = \left( \frac{1}{3r_{ws,SO2}} + \frac{10^{-6}H}{RT_l} + \frac{f_0}{r_{ws,O3}} \right)^{-1} \tag{12}$$

$$r_{soil} = \left( \frac{10^{-4}H}{RT_l r_{soil,SO2}} + \frac{f_0}{r_{soil,O3}} \right)^{-1} \tag{13}$$

Here $r_{stm,H2O}$, the stomatal resistance for water vapor, is obtained from SCADIS module in SOSAA (Zhou et al., 2017). $r_{cut,O_3}$ ($10^5$ s m$^{-1}$), $r_{ws,SO_2}$ (100 s m$^{-1}$), $r_{ws,O_3}$ (2000 s m$^{-1}$), $r_{soil,SO_2}$ (250 s m$^{-1}$), $r_{soil,O_3}$ (400 s m$^{-1}$) are constant values as reference resistances for other gases, here the subscripts $O_3$ and $SO_2$ represent the corresponding resistances of $O_3$ and $SO_2$, respectively. Their values are obtained from Ganzeveld and Lelieveld (1995) and Ganzeveld et al. (1998). $H$ is the Henry's law constant with the unit of M atm$^{-1}$. $f_0$ is the reactivity factor with three values 0, 0.1 and 1, implying non-reactive, slightly-reactive and reactive gases, respectively. $R$ (0.082 atm M$^{-1}$ K$^{-1}$) is the gas constant. $T_l$ is leaf temperature.

### 3.2.2 Henry's law constant ($H$)

The Henry's law constants of 1963 chemical compounds included in the current chemistry scheme have to be acquired to calculate the resistances in Eqs. 10 to 13. First, a compound is searched in the list collected by Sander (2015) (Sander's list). If it is in the list, the most reliable $H$ value for this compound shown in the list is used. Otherwise, the program HENRYWIN (Hine and Mookerjee, 1975; Meylan and Howard, 1991) in the software EPI Suite v4.11 (US EPA) is applied to obtain the $H$ value. The program contains two methods to infer the $H$ values referred to as the group method and the bond method. The performance of these two methods were tested for 4592 compounds in the Sander's list, which indicated that the group method predicted slightly more accurate $H$ values ($R^2 = 0.89$) than the bond method ($R^2 = 0.86$). However, the group method is not available for all the compounds. Hence, the $H$ value derived from the group method is used when available, otherwise, the result from the bond method is used. Finally, the $H$ values of the inorganic compounds nitric acid (HNO$_3$) and hydrogen peroxide (H$_2$O$_2$) are set to $10^{14}$ M atm$^{-1}$ and $5 \times 10^7$ M atm$^{-1}$ (Table S4, Nguyen et al. (2015)).




### 3.2.3 Reactivity factor ($f_0$)

The reactivity factors of all the compounds are determined mainly according to the values and rules suggested by Wesely (1989), Karl et al. (2010) and Knote et al. (2015) (Table 2). The $f_0$ values of sulfur dioxide ($SO_2$), $O_3$, nitrogen dioxide ($NO_2$), nitric oxide (NO), nitric acid ($HNO_3$), hydrogen peroxide ($H_2O_2$), ammonia ($NH_3$), peroxyacetyl nitrate (PAN,

$CH_3C(O)OONO_2$) and nitrous acid (HONO) are retrieved from Table 2 in Wesely (1989). The updated $f_0$ values of formaldehyde, peracetic acid ($CH_3CO_3H$), acetaldehyde, glyoxal (OHCCHO), methylglyoxal ($CH_3C(O)CHO$), glycolaldehyde ($HOCH_2CHO$), 2-hydroxy-3-methylbut-3-enal ($CH_3C(CH_2)CH(CHO)OH$), methanol, ethanol ($CH_3CH_2OH$), acetone, acetol ($CH_2OHC(O)CH_3$), methyl vinyl ketone (MVK, $CH_3C(O)CH{=}CH_2$), methacrolein (MACR, $CH_3C(CH_2)CHO$) and OVOCs with -OOH functional group (ROOH) are proposed by Karl et al. (2010). In addition, the $f_0$ values of OH, $NO_3$, $O(^1D)$, $O(^3P)$, $HO_2$ are set according

to Table S4 in Ashworth et al. (2015). Knote et al. (2015) found that there was no significant difference of semi-volatile organic compounds (SVOCs) deposition when $f_0$ values were set to 0, 0.1 and 1. Hence, they set $f_0$ to 0 for SVOCs, regarding them as non-reactive. Therefore, in this study for the compounds other than those mentioned in Wesely (1989), Karl et al. (2010) and Ashworth et al. (2015), their $f_0$ values are set to 0.

## 3.3 Model setup

### 3.3.1 Meteorology

In order to validate the newly developed gas dry deposition model and then analyze the BVOC exchange processes between the boreal forest canopy and the atmosphere, the model is set up to simulate the time period from July 1st to July 31st in 2010 (Day of year 182 to 212) with the canopy configuration at SMEAR II. The model contains 51 logarithmically-distributed layers from 0 m at soil surface (layer 1) to 3000 m in free troposphere (layer 51). The understory vegetation under ~0.3 m is included

in layer 2 and considered as broadleaved species in the model. Above that the needle-leaved part of dominant coniferous trees are included in layers 3 to 19 within the canopy. The running time step is set to 10 s due to implicit time integration method used in model calculations and the output time step is 30 min.

     The main meteorological diagnostic variables $u$ (eastward wind), $v$ (northward wind), $T$ and $q_v$ (specific humidity) at the upper boundary are constrained by the ERA-Interim reanalysis data obtained from the European Centre for Medium-Range

Weather Forecasts (ECMWF; Dee et al., 2011). The lower boundary is set to non-slip and the measured soil heat flux at SMEAR II are used in surface energy balance calculations. At the canopy top, the long wave radiation provided by the ERA-Interim dataset, as well as the measured downward direct and diffuse global radiation at SMEAR II are used as input. While inside the canopy, three bands of the radiation (long-wave, near-infrared and PAR) at each layer are computed by the meteorology module. A linear interpolation is applied on all the input data to match with the model running time step.





**Table 2.** Reactivity factors ($f_0$) of all the compounds included in the simulation and their references.

| Context name | $f_0$ | Reference |
|---|---|---|
| SO$_2$ | 0 | Wesely (1989) |
| O$_3$ | 1 | Wesely (1989) |
| NO$_2$ | 0.1 | Wesely (1989) |
| NO | 0 | Wesely (1989) |
| HNO$_3$ | 0 | Wesely (1989) |
| H$_2$O$_2$ | 1 | Wesely (1989) |
| NH$_3$ | 0 | Wesely (1989) |
| PAN | 0.1 | Wesely (1989) |
| HONO | 0.1 | Wesely (1989) |
| formaldehyde | 1 | Karl et al. (2010) |
| peracetic acid | 1 | Karl et al. (2010) |
| acetaldehyde | 1 | Karl et al. (2010) |
| glyoxal | 1 | Karl et al. (2010) |
| methylglyoxal | 1 | Karl et al. (2010) |
| glycolaldehyde | 1 | Karl et al. (2010) |
| 2-hydroxy-3-methylbut-3-enal | 1 | Karl et al. (2010) |
| methanol | 1 | Karl et al. (2010) |
| ethanol | 1 | Karl et al. (2010) |
| acetone | 1 | Karl et al. (2010) |
| acetol | 1 | Karl et al. (2010) |
| MVK | 1 | Karl et al. (2010) |
| MACR | 1 | Karl et al. (2010) |
| ROOH | 1 | Karl et al. (2010) |
| OH | 1 | Ashworth et al. (2015) |
| NO$_3$ | 1 | Ashworth et al. (2015) |
| O($^1$D) | 0 | Ashworth et al. (2015) |
| O($^3$P) | 0 | Ashworth et al. (2015) |
| HO$_2$ | 1 | Ashworth et al. (2015) |

### 3.3.2 Chemistry

The chemistry scheme is based on Mogensen et al. (2015). The full MCMv3.2 oxidation paths of methane (CH$_4$), isoprene, MBO, $\alpha$-pinene (C$_{10}$H$_{16}$), $\beta$-pinene (C$_{10}$H$_{16}$), limonene (C$_{10}$H$_{16}$) and $\beta$-caryophyllene (C$_{15}$H$_{24}$) are included with necessary



inorganic reactions. For those emitted BVOCs which are not described by MCM, including 1,8-cineole ($C_{10}H_{18}O$), $\Delta^3$-carene ($C_{10}H_{16}$), other minor monoterpenes (OMT), farnesene ($C_{15}H_{24}$) and other sesquiterpenes (OSQ), their first-order oxidation reactions with OH, $O_3$ and $NO_3$ are added (Atkinson, 1997). In addition, the updated chemical reactions of stabilized Criegee intermediates (sCIs) are also added (Boy et al., 2013). The condensation sinks of sulfuric acid ($H_2SO_4$) and $HNO_3$ are computed according to Kulmala et al. (2001). The measured concentrations of trace gases NO, $NO_2$ ($NO_x$-NO), $SO_2$, CO, $CH_4$, hydrogen ($H_2$) and $O_3$ are used to constrain the model (Mogensen et al., 2015). The initial concentrations of all the other compounds are 0.

### 3.3.3 Emission

The emissions of 15 organic compounds are included in current simulations, which are $\alpha$-pinene, $\beta$-pinene, $\Delta^3$-carene, limonene, 1,8-cineole, OMT, $\beta$-caryophyllene, farnesene, OSQ, isoprene, MBO, methanol, acetaldehyde, acetone, formaldehyde. Their standard emission potentials (SEPs) for July, 2010 at SMEAR II applied in the model and proposed in previous studies are shown in Table 3. It should be noted here that the SEP values in previous studies were obtained during different time periods, in different measurement scales and even by different standardised methods (e.g., Lindfors and Laurila, 2000; Tarvainen et al., 2005; Hakola et al., 2006; Rantala et al., 2015), therefore the selected optimum monthly mean SEPs are within the range of measured SEPs or represent the measured fluxes. Hence, the SEP of total monoterpenes are set to 1227.4 ng g(dw)$^{-1}$ h$^{-1}$ in the range of 838 to 1768.2 ng g(dw)$^{-1}$ h$^{-1}$ (Lindfors and Laurila, 2000; Tarvainen et al., 2005; Hakola et al., 2006; Rantala et al., 2015). Then the SEPs of individual monoterpenes ($\alpha$-pinene, $\beta$-pinene, $\Delta^3$-carene, limonene, 1,8-cineole, OMT) are obtained from their average emission spectra (Bäck et al., 2012). The SEPs of farnesene, $\beta$-caryophyllene and OSQ are set to 45.0 ng g(dw)$^{-1}$ h$^{-1}$, 196.2 ng g(dw)$^{-1}$ h$^{-1}$ within the range of 127 to 385 ng g(dw)$^{-1}$ h$^{-1}$ (Tarvainen et al., 2005; Hakola et al., 2006) and 4.8 ng g(dw)$^{-1}$ h$^{-1}$, respectively. The SEP of total sesquiterpenes is thus 246.0 ng g(dw)$^{-1}$ h$^{-1}$ within the range of 159 to 477 ng g(dw)$^{-1}$ h$^{-1}$ (Hakola et al., 2006). The SEP of MBO is 41.3 ng g(dw)$^{-1}$ h$^{-1}$ lying in the range of 28 to 56 ng g(dw)$^{-1}$ h$^{-1}$ (Tarvainen et al., 2005; Hakola et al., 2006). Since the total SEP of isoprene and MBO are suggested as 445.6 ng g(dw)$^{-1}$ h$^{-1}$ in Rantala et al. (2015), we thus set the SEP of isoprene as 400 ng g(dw)$^{-1}$ h$^{-1}$.

The SEP of methanol is estimated to be ~75 ng m$^{-2}$ s$^{-1}$ by considering both emission and deposition processes for July at SMEAR II in Rantala et al. (2015). Therefore, we use the same value (530.5 ng g(dw)$^{-1}$ h$^{-1}$) after converting the unit to ng g(dw)$^{-1}$ h$^{-1}$ with a biomass of 509 g(dw) m$^{-2}$. For acetone, Janson and de Serves (2001) proposed a value of 870 ± 480 ngC g(dw)$^{-1}$ h$^{-1}$ (1401.7 ± 773.3 ng g(dw)$^{-1}$ h$^{-1}$). Hence we set the SEP of acetone to 974.1 ng g(dw)$^{-1}$ h$^{-1}$ which still lies within the uncertainty range. The SEPs of acetaldehyde and formaldehyde are selected to represent the measured fluxes.

### 3.3.4 Selected compounds

Several representative compounds are selected to analyze the sources and sinks within the canopy for typical BVOCs. (Table 4). Monoterpenes, isoprene, MBO, methanol, acetaldehyde, acetone and formaldehyde are chosen to verify the model by comparing their modeled and measured fluxes above the canopy. These seven compounds along with the sesquiterpenes constitute




**Table 3.** Standard emission potentials (SEP) of selected emitted BVOCs. The values used in SOSAA (monthly mean), the corresponding reference values (average $\pm$ standard deviation) and reference literatures are shown. The last column shows how the reference values are standardised, according to PAR, $T$ or both. The unit of SEP is ng g(dw)$^{-1}$ h$^{-1}$.

| Context name | SOSAA value | Reference value | Reference and remark | Standardization parameters |
|---|---|---|---|---|
| monoterpenes | 1227.4 | 1500$\pm$0 | Lindfors and Laurila (2000) | PAR, $T$ |
| | | 1015$\pm$52 | Tarvainen et al. (2005) | $T$ |
| | | 838$\pm$241, 1106$\pm$466 | Hakola et al. (2006) | $T$ |
| | | 1768.2$\pm$141.5 | Rantala et al. (2015) | PAR, $T$ |
| $\alpha$-pinene | 536.4 | 0.437·SEP(monoterpenes) | Bäck et al. (2012) | |
| $\beta$-pinene | 110.5 | 0.090·SEP(monoterpenes) | Bäck et al. (2012) | |
| $\Delta^3$-pinene | 486.1 | 0.396·SEP(monoterpenes) | Bäck et al. (2012) | |
| limonene | 28.2 | 0.023·SEP(monoterpenes) | Bäck et al. (2012) | |
| 1,8-cineole | 1.2 | 0.001·SEP(monoterpenes) | Bäck et al. (2012) | |
| OMT | 65.1 | 0.053·SEP(monoterpenes) | Bäck et al. (2012) | |
| sesquiterpenes | 246.0 | 477$\pm$131, 159$\pm$51 | Hakola et al. (2006) | $T$ |
| farnesene | 45.0 | | | |
| $\beta$-caryophyllene | 196.2 | 160$\pm$160 | Tarvainen et al. (2005) | $T$ |
| | | 127$\pm$35, 385$\pm$112 | Hakola et al. (2006) | $T$ |
| OSQ | 4.8 | | | |
| isoprene | 400 | 445.6$\pm$28.3 | The reference value referred to the sum of isoprene and MBO (Rantala et al., 2015). | PAR, $T$ |
| MBO | 41.3 | 28$\pm$1 | Tarvainen et al. (2005) | $T$ |
| | | 28$\pm$7, 56$\pm$19 | Hakola et al. (2006) | PAR, $T$ |
| | | 445.6$\pm$28.3 | The reference value referred to the sum of isoprene and MBO (Rantala et al., 2015). | PAR, $T$ |
| methanol | 530.5 | 530.5$\pm$35.4 | Rantala et al. (2015) | $T$ |
| acetone | 974.1 | 1401.7$\pm$773.3 | Janson and de Serves (2001) | $T$ |
| formaldehyde | 530.5 | | | |
| acetaldehyde | 249.8 | | | |

the majority of the emitted organic gases from the ecosystem at SMEAR II. Acetol is further selected as an additional example of a typical carbonyl compound (on top of acetaldehyde, methanol and formaldehyde). Moreover, four increasingly oxi-





dized organic compounds with different carbon chain lengths and chemical functionalities are selected, including ISOP34OOH ($C_5H_{10}O_3$) and ISOP34NO3 ($C_5H_9ONO_3$) both of which are oxidation products of isoprene, pinic acid ($C_9H_{14}O_4$) obtained from $\alpha$-pinene oxidation and BCSOZOH ($C_{15}H_{26}O_5$) produced from $\beta$-caryophyllene oxidation. These compounds were included to be able to simulate the influence of consecutive oxidation and size of the molecule (i.e., changing volatility and

5   Henry's law constant) on the deposition efficiency. They span a range of volatilities and solubilities and thereby have different tendencies to deposit onto surfaces.

**Table 4.** A list of selected featured BVOCs with their Henry's law constants ($H$), the $H$ method references (SE as from Sander (2015), MH as manually set, EB as calculated with bond method by EPI Suite v4.11, EG as calculated with group method by EPI Suite v4.11), the reactivity factors ($f_0$), the $f_0$ references and remarks.

| Context name | $H$ (M atm$^{-1}$) | $H$ reference | $f_0$ | $f_0$ reference | Remark |
|---|---|---|---|---|---|
| $\alpha$-pinene | $3.0 \times 10^{-2}$ | SE | 0 | others | |
| $\beta$-pinene | $1.6 \times 10^{-2}$ | SE | 0 | others | |
| $\Delta^3$-pinene | $1.6 \times 10^{-2}$ | SE | 0 | others | |
| myrcene | $8.9 \times 10^{-2}$ | SE | 0 | others | |
| sabinene | $1.6 \times 10^{-2}$ | SE | 0 | others | |
| ocimene | $3.0 \times 10^{-2}$ | SE | 0 | others | |
| limonene | $4.9 \times 10^{-2}$ | SE | 0 | others | |
| 1,8-cineole | 6.0 | SE | 0 | others | |
| OMT | $2.3 \times 10^{-2}$ | MH | 0 | others | $H = 0.5 \cdot [H(\alpha\text{-pinene}) + H(\beta\text{-pinene})]$ |
| isoprene | $1.3 \times 10^{-2}$ | SE | 0 | others | |
| MBO | 65 | SE | 0 | others | |
| $\beta$-caryophyllene | $1.45 \times 10^{-3}$ | EB | 0 | others | |
| farnesene | 0.102 | EG | 0 | others | |
| OSQ | $1.45 \times 10^{-3}$ | MH | 0 | others | $H = H(\beta\text{-caryophellene})$ |
| formaldehyde | $3.2 \times 10^3$ | SE | 1 | Karl et al. (2010) | |
| methanol | $2.0 \times 10^2$ | SE | 1 | Karl et al. (2010) | |
| acetaldehyde | 13 | SE | 1 | Karl et al. (2010) | |
| acetone | 28 | SE | 1 | Karl et al. (2010) | |
| acetol | $7.8 \times 10^3$ | SE | 1 | Karl et al. (2010) | |
| pinic acid | $1.70 \times 10^9$ | EG | 0 | others | |
| BCSOZOH | $9.09 \times 10^7$ | EB | 0 | others | |
| ISOP34NO3 | $5.05 \times 10^4$ | EB | 0 | others | |
| ISOP34OOH | $1.47 \times 10^6$ | EB | 1 | Karl et al. (2010) | |





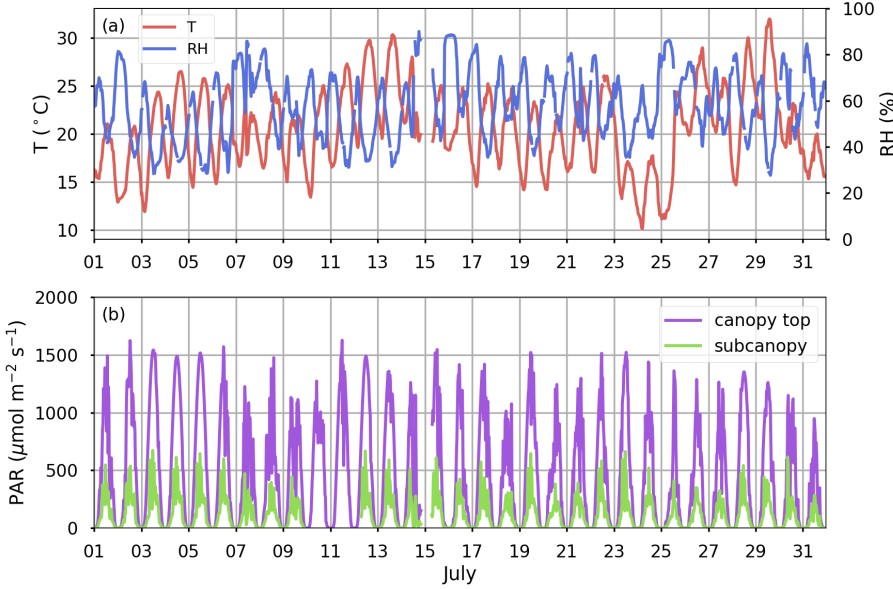

**Figure 1.** Measured monthly time series of (a) air temperature (red), relative humidity (blue), (b) incoming PAR at canopy top (purple) and PAR at subcanopy (green) in July, 2010 at SMEAR II.

## 4 Results and discussion

### 4.1 Micrometeorology

During July, 2010 at SMEAR II, which was a record warm summer in Finland (Williams et al., 2011), the air temperature varied from 10.1 °C to 32.0 °C with a monthly-average of 24.5 °C. The RH showed an opposite diurnal alteration with respect

to air temperature with a mean value of 57.5%, ranging from 27.7% to 90.0% (Fig. 1a). The daytime maximum incoming PAR at the canopy top was larger than 1000 $\mu$mol m$^{-2}$ s$^{-1}$ during the whole month except July 31st, and even reached over 1500 $\mu$mol m$^{-2}$ s$^{-1}$ on nine days. In the sub-canopy (0.6 m) the incoming PAR was only about 1/4 of that at the canopy top, implying apparently slower photochemical reactions happening inside the canopy (Fig. 1b). The accumulated precipitation (liquid water equivalent) of the whole month was 34.64 mm. Hence, overall the month can be described as sunny and dry, with

occasional cloudiness or little precipitation occurring during some of the days.

The simulated and measured July average night and daytime vertical profiles of horizontal wind speed, air temperature, as well as diurnal cycles in the friction velocity at 23 m and in-canopy average RH are shown in Fig. 2. The wind speed, vertical potential temperature gradient and friction velocity mainly reflect (and depend on) the vertical mixing conditions inside and above the canopy, which is essential for estimating the overall BVOC exchange inside the canopy. During daytime

(sun elevation angle is larger than 10°), the observed wind speed shows a large decrease from 3.4 m s$^{-1}$ above the canopy to 0.9 m s$^{-1}$ deeper inside the canopy due to canopy drag. The nighttime (sun elevation angle is smaller than 0°) profile shows a





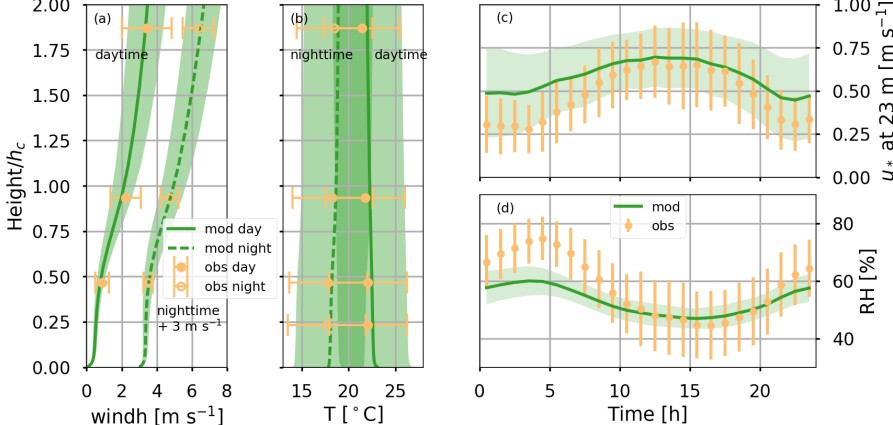

**Figure 2.** Modeled (green solid line for daytime, green dashed line for nighttime) and measured (yellow solid circle for daytime, yellow empty circle for nighttime) profiles of (a) horizontal wind speed (windh) and (b) air temperature ($T$). Nighttime values are shifted by 3 m s$^{-1}$ for wind for clarity of presentation. The ranges of $\pm 1$ SD (standard deviation) of modeled and measured data are marked as shades and error bars. The height is normalised by canopy height ($h_c$). The monthly-averaged diurnal cycles of modeled (green line) and measured (yellow dots) (c) friction velocity ($u_*$) at 23 m and (d) mean RH inside the canopoy are also plotted. The ranges of $\pm 1$ SD of modeled and measured data are marked as shades and vertical lines.

similar pattern (Fig. 2a). Here we focus on the meteorological conditions below about $2h_c$ (36 m), where the air pressure and density can be considered the same as that at the ground level. Hence, the air temperature can be assumed to be the potential temperature within this vertical range. The observed vertical profile of temperature shows a negative upward gradient within and above the canopy during daytime, indicating the occurence of unstable condition which facilitates vertical mixing within

and above the canopy. In contrast, the positive upward gradient in temperature profile implies an inhibition of turbulence motions during nighttime (Fig. 2b). The horizontal wind speed and the temperature are well simulated by the model (Figs. 2a and b). The measured diurnal cycle of friction velocity at canopy top indicates more production of turbulence at daytime compared to that at nighttime due to buoancy term (Fig. 2c). The model overestimates the nighttime friction velocity, which may imply not only an excessive mixing between the canopy and overlaying inversion layer for nocturnal BVOC exchange but

also a possible misrepresentation of other drivers of BVOC sources and sinks such as moisture conditions as discussed below. The observed monthly-averaged RH values exceed 70% from about 02:00LT to 06:00LT, indicating the occuring of wet skin on the leaf surface (Altimir et al., 2006; Zhou et al., 2017). However, the model underestimates the RH values during night and in the early morning, resulting in simulated RH values generally less than 70%, on average actually not larger than 60%, during the simulation period (Fig. 2d). Therefore, the observed RH values inside the canopy were used to parametrise $f_{wet}$

when calculating the deposition velocity to represent a more realistic leaf wetness condition, also since this leaf wetness plays a potentially important role in BVOC exchange as we demonstrate in further details below in Section 4.4.



## 4.2 Model validation

The current version of SOSAA with similar setup has been applied and verified in Zhou et al. (2017), hence here we only show the comparisons of simulated and observed parameters which are relevant for BVOC exchange as presented in this study.

### 4.2.1 Energy fluxes

The simulated and measured monthly-averaged diurnal cycles of energy fluxes for the canopy-soil ecosystem are compared in order to verify the modeled micrometeorology with a focus on the radiation and energy balance (Fig. 3). During daytime, e.g. at 12:30LT, the measured downward net radiation (-414 W m$^{-2}$) is approximately balanced by sensible heat flux (200 W m$^{-2}$), latent heat flux (190 W m$^{-2}$) and a small soil heat flux (25 W m$^{-2}$) from the ecosystem into the soil. During nighttime, e.g. at 01:30LT, the net upward long wave radiation (44 W m$^{-2}$) along with minor latent heat flux (4 W m$^{-2}$) is partly compensated by a downward sensible heat flux (-27 W m$^{-2}$), resulting in an overall nocturnal decrease of the canopy temperature and onset of a stable inversion at the canopy top.

Although the model underestimates the monthly-averaged diurnal sensible heat flux from 11:00LT to 20:00LT by a maximum of 76 W m$^{-2}$, the simulation results of energy fluxes show an acceptable agreement with the measurements. Moreover, the promising agreement between modeled and measured latent heat flux indicates a realistic representation of the water vapor exchange between the air and the ecosystem, which hints at a realistic representation of stomatal resistance essential for the representation of stomatal removal of the chemical compounds in the model.

### 4.2.2 BVOC fluxes

The BVOC emissions in SOSAA are simulated by MEGAN with prescribed standard emission potentials. The modeled emissions of monoterpenes were evaluated by Smolander et al. (2014) via comparisons between simulated and measured fluxes and concentrations for June 2007 at SMEAR II. In this study, the simulated fluxes at the canopy top for six different emitted compounds or groups (including monoterpenes, isoprene+MBO, methanol, acetaldehyde, acetone, formaldehyde) are compared with the measurements.

Figure 4 shows the modeled and measured monthly mean diurnal cycles in BVOC fluxes at the canopy top. The measured fluxes of monoterpenes, isoprene+MBO, methanol, acetaldehyde and acetone show a similar diurnal pattern mainly following the diurnal patterns of emission intensities (Figs. 4a-e). During daytime, the fluxes of these BVOCs increase continuously and reach a maximum at around 14:00LT in the afternoon. The observed nighttime upward fluxes of these BVOCs, except monoterpenes whose emission is strongly regulated by temperature instead of light, are close to zero when both the emission and the turbulence are small. For methanol and acetaldehyde, the measured fluxes can be downward at nighttime and in the early morning, due to gas dry deposition showing bidirectional fluxes (Figs. 4c and d). Schallhart et al. (2016) also observed considerable downward flux of methanol from 01:00LT to 08:00LT over a Mediterranean oak-hornbeam forest, and proposed that this was due to deposition under the presence of dew. The measured monthly-averaged diurnal flux for formaldehyde is mostly downward and does not show an apparent diurnal pattern. The observed large range in formaldehyde fluxes also indicate





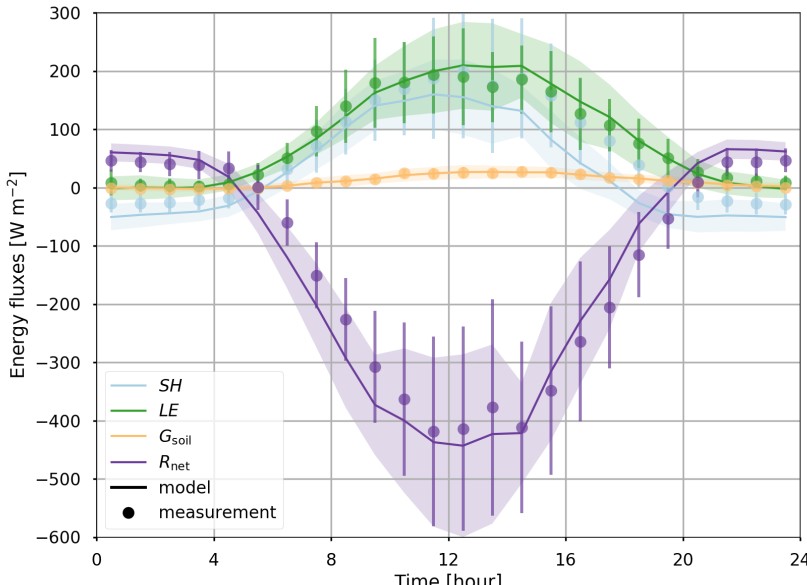

**Figure 3.** The modeled (solid lines) and measured (points) monthly-averaged diurnal cycles of sensible heat flux ($SH$, blue), latent heat flux ($LE$, green), soil heat flux ($G_{soil}$, yellow) and upward net radiation ($R_{net}$, purple, the observed $R_{net}$ is at 67 m). The ranges of $\pm 1$ standard deviation for modeled and measured data are marked by shaded areas and vertical lines, respectively.

that, although the deposition may play a significant role in its exchange processes, other effects, e.g. emission and chemical reactions, might provide a comparable contribution to the overall formaldehyde source-sink balance (Fig. 4f).

The diurnal variations of fluxes for monoterpenes, isoprene+MBO, methanol and acetaldehyde are well represented by the model. Although for isoprene+MBO the monthly-averaged daytime flux is underestimated up to about 0.025 $\mu$g m$^{-2}$ s$^{-1}$

(~65.0%) at 17:30LT, the values are still in the range of the measurement uncertainties (Fig. 4b). For acetone, the model underestimates the upward flux in the morning and shows a dominant downward flux around 04:00LT which is not seen in the observations, implying a potential overestimation of the role of deposition or a missing source in canopy exchange of acetone. In contrast, the model overestimates the upward flux from ~10:00LT to ~16:00LT at daytime probably due to excessive sources (Fig. 4e). The model overestimates the downward flux of formaldehyde in the morning from ~04:00LT to ~12:00LT, and

does not capture the observed abrupt increase in this downward flux between 12:00LT and 16:00LT. However, considering the large uncertainties of measurements of formaldehyde flux as mentioned in Sec. 2.2.2, the differences between modeled and observed diurnal variation of formaldehyde flux do not indicate a poor performance regarding the simulations of formaldehyde sources, sinks and exchange (Fig. 4f). In summary, considering the 3-D nature of the actual observation conditions and the resulting uncertainties introduced in such comparison of a 1-D model results with measurements, there seems to be a good

correspondence between simulated and observed diurnal cycles in BVOC exchange fluxes.





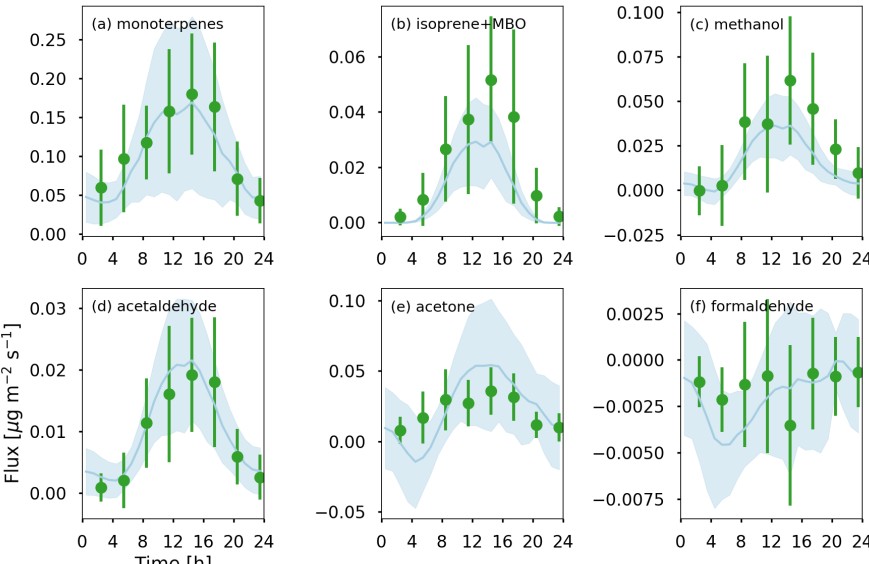

**Figure 4.** Modeled (blue lines) and measured (green points) monthly-averaged diurnal cycles of fluxes for (a) monoterpenes, (b) iso-prene+MBO, (c) methanol, (d) acetaldehyde, (e) acetone and (f) formaldehyde at the canopy top. The ranges of $\pm 1$ standard deviation for modeled and measured data are marked by shaded areas and vertical lines, respectively.

## 4.3 Overview of in-canopy sources and sinks

The simulated monthly-averaged relative contributions of individual in-canopy sources and sinks ($\overline{Q}_{rel,n}^{\Delta,h_c}$, Eq. A10) during the whole day, the daytime and the nighttime are shown in Fig. 5. Figure 5a shows the monthly-averaged relative contributions by emissions, dry deposition, chemistry and turbulent transport in the overall concentration changes during the whole day. For those compounds which are emitted from the canopy, emission is always a significant source within the canopy. However, the sink terms are different for individual gases and we can distinguish three contrasting exchange regimes. First, ~86% of the emitted monoterpenes and ~93 % of isoprene+MBO are transported out of the canopy. Secondly, for the emitted sesquiterpenes, ~70% are removed by chemical oxidation within the canopy due to their very high reactivity, and only ~29% escape the canopy. This result is comparable with the estimation by Rinne et al. (2007), which showed that the fraction of the emitted sesquiterpenes being able to reach the measurement height (22 m) at the same site was about 30 – 40% in July. Rinne et al. (2012) presented a slightly higher ratio between above canopy flux and canopy emission of $\beta$-caryophyllene, which was about 50% during daytime and below 40% during nighttime. Thirdly, dry deposition plays a significant or even dominant role among the removal processes and contributes about 80%, 35%, 100%, 100% to the total sinks for methanol, acetaldehyde, acetone and formaldehyde within the canopy, respectively. Hence their fluxes are bidirectional in the simulation (Figs. 4c-f). In the case of acetone and formaldehyde, the dry deposition sinks exceed the emission sources by about 20% and > 170%, respectively, and where the net canopy sink of these compounds is partly compensated by downward turbulent transport.



During daytime, which lasts about 18 hours in July, the relative contributions by chemistry, deposition and mixing to the overall concentration changes and fluxes for emitted gases change very little compared to the whole-day average and the maximum difference of monthly mean $\overline{Q}_{rel,n}^{\Delta,h_c}$ is less than 0.06 (Fig. 5b). At nighttime, the mass balance patterns for emitted compounds still keep similar except for isoprene+MBO which changes dramatically since the emission reduces a lot due to lack of light (Fig. 5c). Consequently, inside the canopy the source is only ~51% of the sink for isoprene+MBO, which implies an imbalance and thus concentration reduction during nighttime. However, the absolute nighttime concentration change for isoprene+MBO is nearly zero as we will discuss below in Section 4.4.

In general, all the selected non-emitted gases which are chemically produced from the emitted precursor gases are removed by deposition predominantly showing downward fluxes (Fig. 5a). However, their chemistry contribution to total sources vary in a wide range from ~3% (acetol) to ~70% (ISOP34NO3) implying a complicated relation between the vertical distribution of their precursors, the subsequent oxidation reactions and their own deposition processes. According to the monthly-average relative contribution of chemistry ($\overline{Q}_{rel,chem}^{\Delta,h_c}$) during the whole-day, the non-emitted gases can be divided into two categories, one with $\left|\overline{Q}_{rel,chem}^{\Delta,h_c}\right| < 0.25$ and the other with $\left|\overline{Q}_{rel,chem}^{\Delta,h_c}\right| \geq 0.25$.

For the first category, the modeled relative contribution by deposition is much larger than the in-canopy chemical production which is ~3%, ~11% and ~18% of the deposited gases for acetol, pinic acid and BCSOZOH, respectively. The chemistry contributions of them change less than 0.05 at daytime (Fig. 5b) and less than 0.10 at nighttime (Fig. 5c) compared to the whole-day average. While in the second category, the oxidation of isoprene initiated by OH can produce ~33% and ~70% of the lost gases for ISOP34OOH and ISOP34NO3 inside the canopy (Fig. 5a). At daytime, their relative contributions of chemistry increase to ~0.42 and ~0.86 due to higher concentration of OH (Fig. 5b). At nighttime, in contrast, they are even destroyed by chemical reactions with $O_3$ given the low OH concentrations (Fig. 5c).

Therefore the selected BVOCs are finally classified into five categories: Cemis in which emitted gases are mostly transported out of the canopy (monoterpenes, isoprene+MBO), Cemis-chem in which emitted gases are removed significantly by chemistry (sesquiterpenes), Cemis-depo in which emitted gases are removed by a significant deposition contribution (acetaldehyde, methanol, acetone, formaldehyde), Cdepo in which the gases are mostly deposited (acetol, pinic acid, BCSOZOH), and finally Cchem-depo in which the chemical production can compensate a significant portion of deposition sink (ISOP34OOH, ISOP34NO3).

This classification also implies that for the gases in categories Cemis-chem and Cemis-depo, it is difficult to determine the actual emission rates only with canopy-top or surface layer flux measurements, since the actual emissions can be significantly reduced by chemical reactions and dry deposition processes. For example, the lack of observations on the latter process implies that we rely to a large extent on parametrisations such as the one proposed by Wesely (1989).

## 4.4 Diurnal cycles of in-canopy sources and sinks

The monthly-averaged diurnal cycles in the source and sink terms $\overline{Q}_n^{\Delta,h_c}$ (Eq. A8) and their relative contributions ($\overline{Q}_{rel,n}^{\Delta,h_c}$) for selected BVOCs are shown in Figs. 6 and 7. All of the $\overline{Q}_{emis}^{\Delta,h_c}$ of the emitted gases are following the diurnal patterns of the temperature and incoming PAR, which keep minimum values during nighttime and reach maximum in the afternoon at





~14:00LT. Among them, the emission of isoprene+MBO strongly depends on the light compared to other compounds, hence its $\overline{Q}_{emis}^{\Delta,h_c}$ is zero during night.

For the category Cemis, the $\overline{Q}_{turb}^{\Delta,h_c}$ term expressing the role of turbulent transport in concentration tendencies show an approximately opposite diurnal pattern compared to $\overline{Q}_{emis}^{\Delta,h_c}$, implying that most of the emitted gases are transported out of the canopy throughout the whole day (Figs. 6a and b). Although the relative contribution of monoterpene emission is about 1.00 during the whole day, the absolute value is altering, e.g. the mean nighttime $\overline{Q}_{emis}^{\Delta,h_c}$ (8.4 $\mu$g m$^{-3}$ h$^{-1}$) is about 58% of mean daytime value (14.5 $\mu$g m$^{-3}$ h$^{-1}$) (Fig. 6a). For isoprene+MBO, there is no nighttime emission of isoprene, hence the nighttime $\overline{Q}_{emis}^{\Delta,h_c}$ is expressing only the contribution by MBO and is much smaller than the daytime emissions from isoprene and MBO. Therefore, although the relative contributions from chemistry and deposition for isoprene+MBO are 100% all together, their absolute contributions to the overall concentration changes are negligible. The chemical loss for monoterpenes and isoprene+MBO is important throughout the boundary layer, but inside the canopy the monthly-averaged chemical destruction tendency $\overline{Q}_{chem}^{\Delta,h_c}$ is only 5 – 10% of $\overline{Q}_{emis}^{\Delta,h_c}$ (Figs. 5a and 6b).

Sesquiterpenes, which belong to the category Cemis-chem, are efficiently destroyed by chemical reactions with O$_3$ within the canopy. Consequently, the diurnal pattern in $\overline{Q}_{chem}^{\Delta,h_c}$ for sesquiterpenes corresponds to the diurnal variations of the O$_3$ concentration. However, the relative contributions of different source and sink terms only change little during the whole day (Fig. 6c).

In the third category Cemis-depo, diurnal changes in the deposition process, e.g., due to changes in mixing conditions, stomatal opening and leaf/needle surface properties, can result in weak or downward fluxes at the canopy top in the morning when the emission is weak (Figs. 6d-f). For formaldehyde, the average absolute value of $\overline{Q}_{depo}^{\Delta,h_c}$ is about 0.90 $\mu$g m$^{-3}$ h$^{-1}$ larger than $\overline{Q}_{emis}^{\Delta,h_c}$, resulting in a downward turbulent flux at the canopy top during the whole day (Fig. 7a). The daily variation reflected by the occurrence of bi-directional fluxes also indicates the difficulty of measuring the actual emission rates of those compounds.

When the turbulent transport and the dry deposition terms are the only source and sink within the canopy, e.g. in the category Cdepo, only downward flux can be observed (Figs. 7b-d). For pinic acid and BCSOZOH, which have very high $H$ values ($1.70 \times 10^9$ M atm$^{-1}$ and $9.09 \times 10^7$ M atm$^{-1}$), the absolute values of $\overline{Q}_{depo}^{\Delta,h_c}$ have mid-night peaks due to higher RH which results in larger wet skin fraction on leaf surface and thus facilitate the deposition of soluble gases onto the leaf surface.

For the category Cchem-depo, the daytime chemical production plays a significant or dominant role in the concentration variations, because the oxidation products ISOP34NO3 and ISOP34OOH are produced from a chain of chemical reactions starting with isoprene oxidation during the daytime (Figs. 7e and f). For ISOP34NO3, $\overline{Q}_{chem}^{\Delta,h_c}$ is even larger than $\overline{Q}_{depo}^{\Delta,h_c}$ at noon, causing weak upward fluxes over the canopy, whereas for ISOP34OOH, the deposition sink is always larger than the chemical production.



### 4.5 Vertical profiles of in-canopy sources and sinks

In order to investigate how different source and sink terms are distributed inside the canopy, the monthly-averaged vertical profiles of $\overline{Q}_n^\Delta$ (Eq. A6) for all the selected compounds are plotted in Figs. 8 and 9. Here the $\overline{Q}_n^\Delta$ values at each layer are weighted by $\Delta z_i/h_c$ ($i$ is the layer index) to represent layer-specific actual contributions to monthly-averaged $\overline{Q}_n^{\Delta,h_c}$.

For all emitted compounds, the vertical distributions of emission source are approximately following the LAD profile with an upward shifting during the whole day, which implies that PAR and leaf temperature play a comparable role in emission rates besides the LAD. Due to strong PAR-dependent emissions, the maximum value of $\overline{Q}_{emis}^\Delta$ for isoprene+MBO locates at ~16 m, which is higher than that of other emitted compounds whose emissions are both PAR and temperature dependent (Figs. 8a–f and 9a). This results from the effect of relatively fast attenuation of PAR inside the canopy compared to the effect of vertical

temperature gradient (Figs. 1b and 2b).

In fact, the vertical distributions of both PAR and leaf temperature depends on the LAD profile which affects the incoming solar radiation. However, due to turbulent mixing, the air temperature distribution is more homogenous inside the canopy also reflected by the relative small vertical gradient in leaf temperature. In contrast, PAR is attenuated within the canopy only as a function of LAD and therefore has larger vertical gradient.

For the BVOCs in categories Cemis-depo, Cdepo, Cchem-depo, dry deposition is significant and even becomes the only dominant sink term for the non-emitted gases. The dry deposition rate above the soil layer is mainly determined by the LAD at each layer inside the canopy. Therefore, the dry deposition follows the vertical profile of LAD. Besides the deposition onto vegetation surface, soil deposition provides an important sink similar to $O_3$ for which the estimated soil deposition sink removes about 36% of all the $O_3$ removed by the boreal forest (Zhou et al., 2017). For BVOCs with significant dry deposition sinks,

the contribution of daily average soil deposition (including deposition onto understory vegetation) to the total deposition varies from 11% (pinic acid) to 61% (ISOP34OOH). Without considering the soil deposition, a majority portion of sources and sinks are located above a height which is about 8 m for monoterpenes, isoprene+MBO, sesquiterpenes, acetaldehyde, methanol and acetone (Figs. 8a-f), and about 4 m for formaldehyde, acetol, pinic acid, BCSOZOH, ISOP34NO3 and ISOP34OOH (Figs. 9a-f). Therefore, below 8 or 4 m depending on specific compounds, the contributions of $\overline{Q}_{emis}^\Delta$, $\overline{Q}_{chem}^\Delta$ and $\overline{Q}_{turb}^\Delta$ can be

neglected. This is also true for $\overline{Q}_{depo}^\Delta$ for the BVOCs with very weak soil deposition, e.g. monoterpenes, isoprene+MBO and sesquiterpenes (Figs. 8a-c).

The vertical profiles of the monthly-averaged total concentration tendencies $\overline{Q}_n^\Delta$ for selected gases, except isoprene+MBO, ISOP34NO3 and ISOP34OOH, only change the magnitude during daytime and nighttime instead of profile patterns (Figs. 8g, 8i-l, m, o-r, 9g-j, m-p). At nighttime, the dry deposition is as important as the emission for isoprene+MBO within the canopy

(Fig. 8n), however, their absolute contributions are too small compared to that at daytime as can be also seen in the diurnal cycle (Fig. 6b). For the isoprene oxidation products ISOP34NO3 and ISOP34OOH, the deposition is compensated by the downward turbulent fluxes without the chemical production during the nighttime, resulting in obvious net removal of the gases throughout the canopy (Figs. 9q and r). Moreover, at the canopy top and close to the surface, these compounds exhibit clear imbalance between production and sink terms, however the imbalance does not really affect the concentration change inside the canopy




since the absolute in-canopy source and sink terms are all close to zero. During daytime, chemical productions of these two BVOCs, which are maximum at the canopy top and decrease inside the canopy, are larger than the deposition sinks above ~14 m (Figs. 9k and l). Thus the extra produced gases at these levels inside the canopy can then be transported to deeper inside the canopy, causing $\overline{Q}^{\Delta}_{turb}$ changing the sign at ~14 m. This phenomena of changing sign of $\overline{Q}^{\Delta}_{turb}$ inside the canopy can also be seen for formaldehyde at both daytime and nighttime (Fig. 9a, g and m). In this case, $\overline{Q}^{\Delta}_{emis}$ is comparable with $\overline{Q}^{\Delta}_{depo}$ but its peak position is higher than that of $\overline{Q}^{\Delta}_{depo}$.

## 5 Summary

Based on the $O_3$ dry deposition model developed in Zhou et al. (2017), a new multi-layer gas dry deposition model extended from Wesely (1989) and Nguyen et al. (2015) has been implemented into the 1D chemical transport model SOSAA. This model enables the calculation of dry deposition processes within a forest canopy for thousands of different gas compounds included in the chemistry scheme. Furthermore, along with the emission and chemistry modules in SOSAA, this new model has been used to analyze individual sources and sinks of 12 selected BVOCs within a boreal forest canopy at SMEAR II in July, 2010, including emissions, chemical production and loss, dry deposition removal and turbulent transport.

In this model, the Henry's law constants are used to calculate parametrised resistances of all the compounds instead of effective Henry's law constants according to the suggestion by Nguyen et al. (2015). The values are obtained from a series of sources in the following order: the experiment data collected in Sander (2015), computed by EPI Suite with group contribution method and computed by EPI Suite with bond contribution method. In addition, the reactivity factors are set based on the values and rules listed in Wesely (1989), Karl et al. (2010), Ashworth et al. (2015) and Knote et al. (2015) (Table 2). With the appropriate setup of standard emission potentials for emitted gases according to previous studies (Table 3), the simulated fluxes at the canopy top for monoterpenes, isoprene+MBO, methanol, acetaldehyde, acetone and formaldehyde agree well with the observed data considering the uncertainties of the measurements (Fig. 4).

The model results of the monthly-averaged $\overline{Q}^{\Delta,h_c}_n$ show that, inside the canopy emission is always the dominant source term for emitted gases of the investigated species except formaldehyde for which the contribution of turbulent transport is larger than emission (Fig. 5a). This indicates that the chemical reactions occurring within the atmospheric boundary layer or other sources such as advection or air masses from source regions can also affect the concentration tendency within the canopy for specific gas species. Moreover, ~86% of the emitted monoterpenes and ~93% of emitted isoprene+MBO are ventilated out of the canopy, while only ~29% of emitted sesquiterpenes are transported away with ~70% consumed by oxidation reactions within the canopy, which is comparable with previous studies. The other four compounds (acetaldehyde, methanol, acetone and formaldehyde) with significant dry deposition sinks can have either mean upward (acetaldehyde and methanol) or downward (acetone, formaldehyde) fluxes at the canopy top depending on the magnitudes of $\overline{Q}^{\Delta,h_c}_{emis}$ and $\overline{Q}^{\Delta,h_c}_{depo}$ (Fig. 5a). Therefore, the overall in-canopy interactions can result in the occurrence of canopy-scale bi-directional exchange. For the selected non-emitted gases, dry deposition is the only dominant sink term, resulting in predominant downward fluxes. ISOP34OOH and ISOP34NO3 are significantly chemically produced inside the canopy, compensating ~33% and ~70% of the deposition loss.





In contrast, at nighttime, they are removed by chemical reactions although the contributions are less than 6% of the deposition loss (Fig. 5c). According to the significance of different source and sink terms, the selected BVOCs can be classified into five categories: Cemis (monoterpenes, isoprene+MBO), Cemis-chem (sesquiterpenes), Cemis-depo (acetaldehyde, methanol, acetone, formaldehyde), Cdepo (acetol, pinic acid, BCSOZOH), Cchem-depo (ISOP34OOH, ISOP34NO3), where the subscripts

represent the significant terms. These findings on the different exchange regimes also further stress the need of the application of canopy exchange modelling system rather than applying the still commonly applied big-leaf representation without considering these interactions between chemistry, emissions and deposition.

The monthly-averaged diurnal variations of in-canopy sources and sinks have also been analyzed. First, the emissions follow the temperature and PAR diurnal patterns, and the nighttime emission values are about 30 – 50% of that at daytime

for monoterpenes, sesquiterpenes, acetaldehyde, methanol, acetone, formaldehyde (Figs. 6a, 6c-f, 7a), while the nighttime emission contribution is approximately zero for isoprene+MBO, which is mostly controlled by light (Fig. 6b). Secondly, the chemical production and loss depend on specific species which usually peak around noon (sesquiterpenes, ISOP34NO3, ISOP34OOH) (Figs. 6c, 7e and f). However, several gases, whose production does not mainly rely on photochemistry, may have peaks of chemical production at mid-night (pinic acid, BCSOZOH) (Figs. 7c and d). In addition, the chemical production

of ISOP34NO3 at noon is even larger than its deposition sink, causing slightly upward flux at the canopy top (Fig. 7e). Thirdly, for the gases in the category Cemis-depo, the turbulent fluxes at the canopy top are bidirectional depending on the intensities of emission sources and dry deposition sinks. The difference between dry deposition and emission fluxes is usually largest in the morning, resulting in downward fluxes, e.g. for methanol and acetone (Figs. 6e and f).

The vertical distributions of emission sources are peaking at ~16 m for isoprene+MBO (Figs. 8b, h and n) and ~14 m for

other emitted gases (Figs 8a, c-f, g, i-l, m, o-r, 9a, g and m). These peaks of emissions are located higher than the level where we simulate the maximum contribution by dry deposition which is located at ~10 m consistent with the LAD profile. Nearly all the source and sink terms except soil deposition seem to show their largest contributions to the overall concentration tendencies above 8 m for monoterpenes, isoprene+MBO, sesquiterpenes, acetaldehyde, methanol and acetone, and about 4 m for formaldehyde, acetol, pinic acid, BCSOZOH, ISOP34NO3 and ISOP34OOH (Figs. 8 and 9). The soil deposition is significant

for the gases in categories Cemis-depo, Cdepo, Cchem-depo, contributing 11% - 61% to the total deposition inside the canopy. For ISOP34NO3 and ISOP34OOH, which are produced by chemical reactions, the largest contributions by deposition to the total concentration changes are found near the canopy top and decreases going down through the canopy.

This study has also provided a method to quantify the proportion of deposition sinks for various BVOCs which can be applied in large-scale models in future. On the basis of the analysis of 12 selected BVOCs and groups of BVOCs in this study, a large

amount of other compounds with similar properties can be represented by being classified into the five categories mentioned above. For example, OVOCs most likely belong to categories Cdepo and Cchem-depo, which indicates that dry deposition can not be neglected when their sources and sinks are investigated. In addition, the categories Cemis-chem and Cemis-depo imply that the simulation of individual processes is necessary to help further analyse the measured emission data of such gases, and thus obtain a more accurate estimation of BVOC exchanges. This study has shown that dry deposition of oxidation

products of precursor gases as well as other BVOCs could be a potentially important feature of improving our understanding





and quantification of BVOC exchange. However, such assessments are largely limited by available observations that could further corroborate the correctness of the simulated deposition processes as presented in this study. In addition, this study stresses the necessity of applying a canopy exchange modeling system for a detailed analysis of BVOCs exchange regimes within and above a boreal forest canopy, instead of applying a big-leaf representation without considering the interactions

5   between chemistry, emissions and deposition.





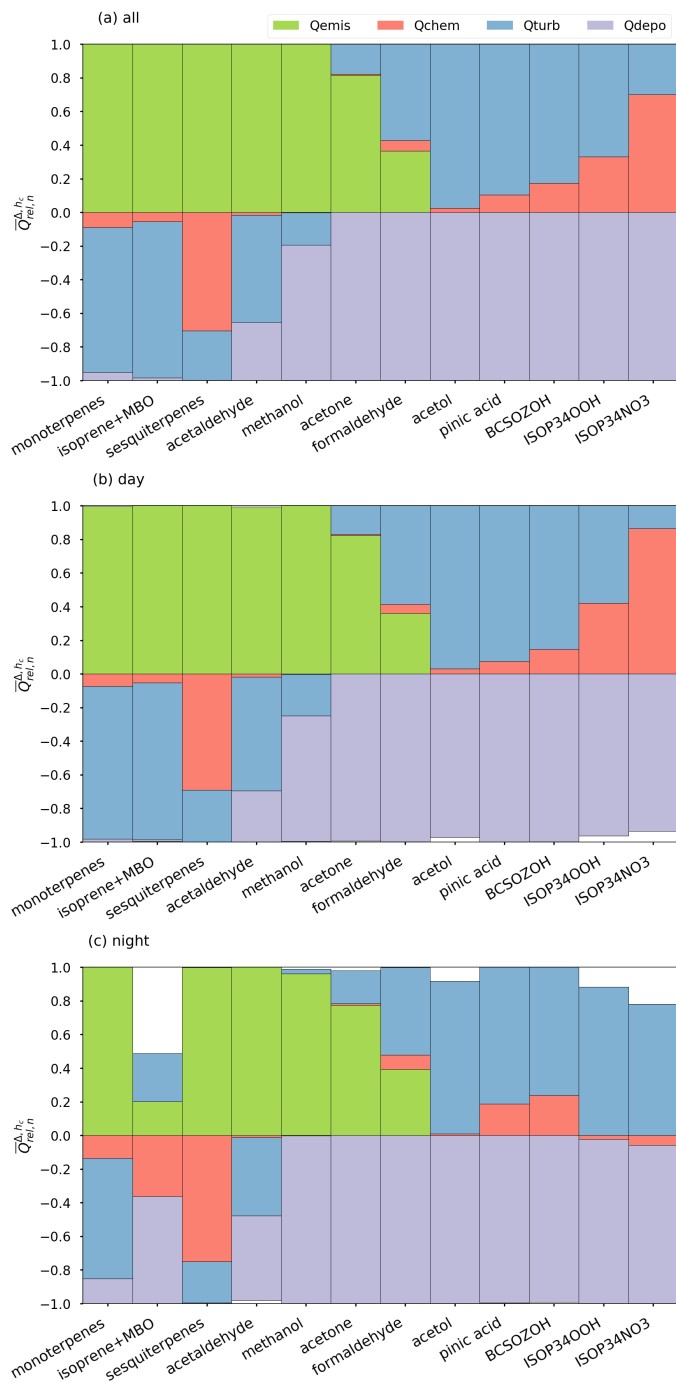

**Figure 5.** Monthly-averaged relative contributions of in-canopy sources and sinks ($\overline{Q}_{rel,n}^{\Delta,h_c}$), including gas emissions (emis, green), net chemical production and loss (chem, red), turbulent transport (turb, blue) and gas dry deposition (depo, purple) for selected BVOCs during (a) the whole month, (b) daytime and (c) nighttime.





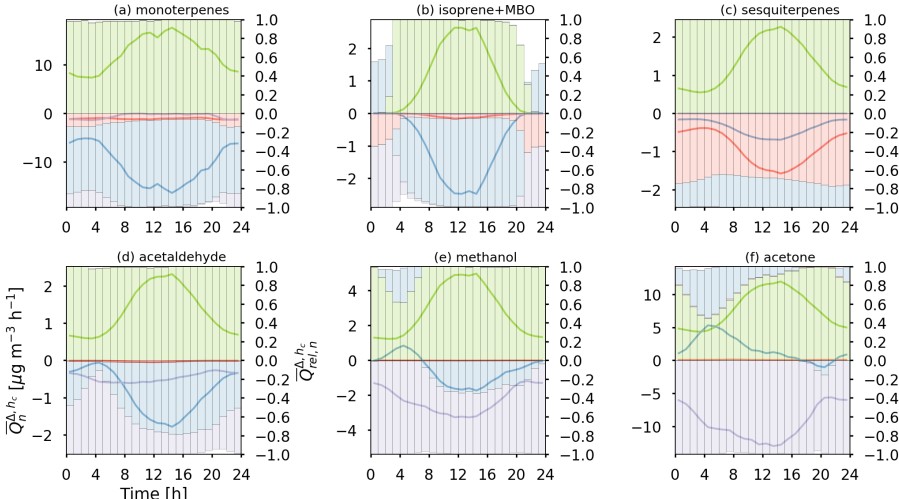

**Figure 6.** Modeled monthly-averaged diurnal cycle of $\overline{Q}_n^{\Delta,h_c}$ (solid lines) and the relative contributions $\overline{Q}_{rel,n}^{\Delta,h_c}$ (bars) of gas emissions (green), net chemical production and loss (red), turbulent transport (blue) and gas dry deposition (purple) within the canopy for (a) monoterpenes, (b) isoprene+MBO, (c) acetaldehyde, (d) sesquiterpenes, (e) methanol and (f) acetone.

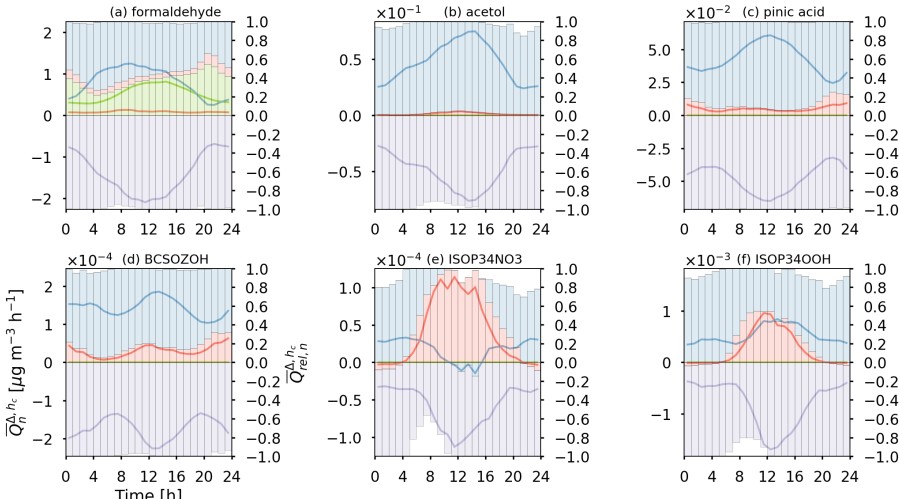

**Figure 7.** The same as Fig. 6 except for (a) formaldehyde, (b) acetol, (c) pinic acid, (d) BCSOZOH, (e) ISOP34NO3 and (f) ISOP34OOH.




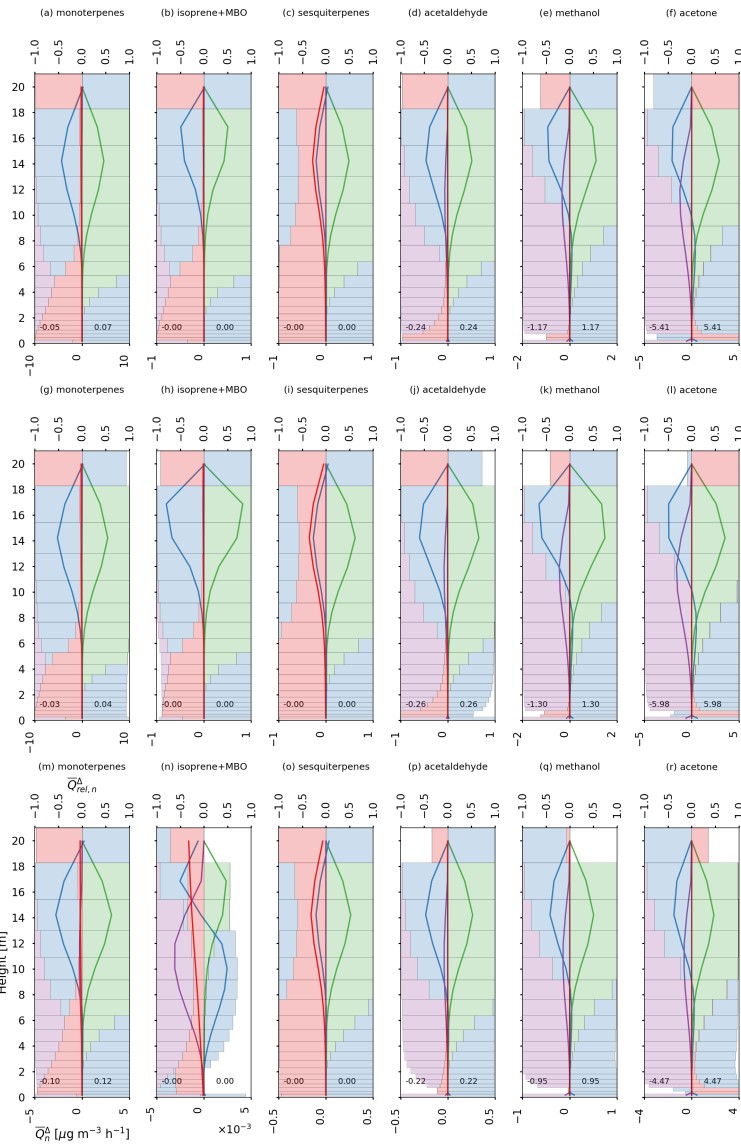

**Figure 8.** Modeled monthly-averaged vertical profiles of weighted $\overline{Q}_n^\Delta$ (solid lines) and the relative contributions $\overline{Q}_{rel,n}^\Delta$ (bars) of gas emissions (green), net chemical production and loss (red), turbulent transport (blue) and gas dry deposition (purple) within the canopy for (a) monoterpenes, (b) isoprene+MBO, (c) acetaldehyde, (d) sesquiterpenes, (e) methanol and (f) acetone. The second panels (g) to (l) and the third panels (m) to (r) are for the same compounds but the average is done for daytime and nighttime, respectively. The values of weighted $\overline{Q}_n^\Delta$ at surface layer are divided by 10 for clarity. The original values at surface layer for deposition (left) and transport (right) are shown as float numbers at the bottom for each plot.

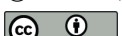



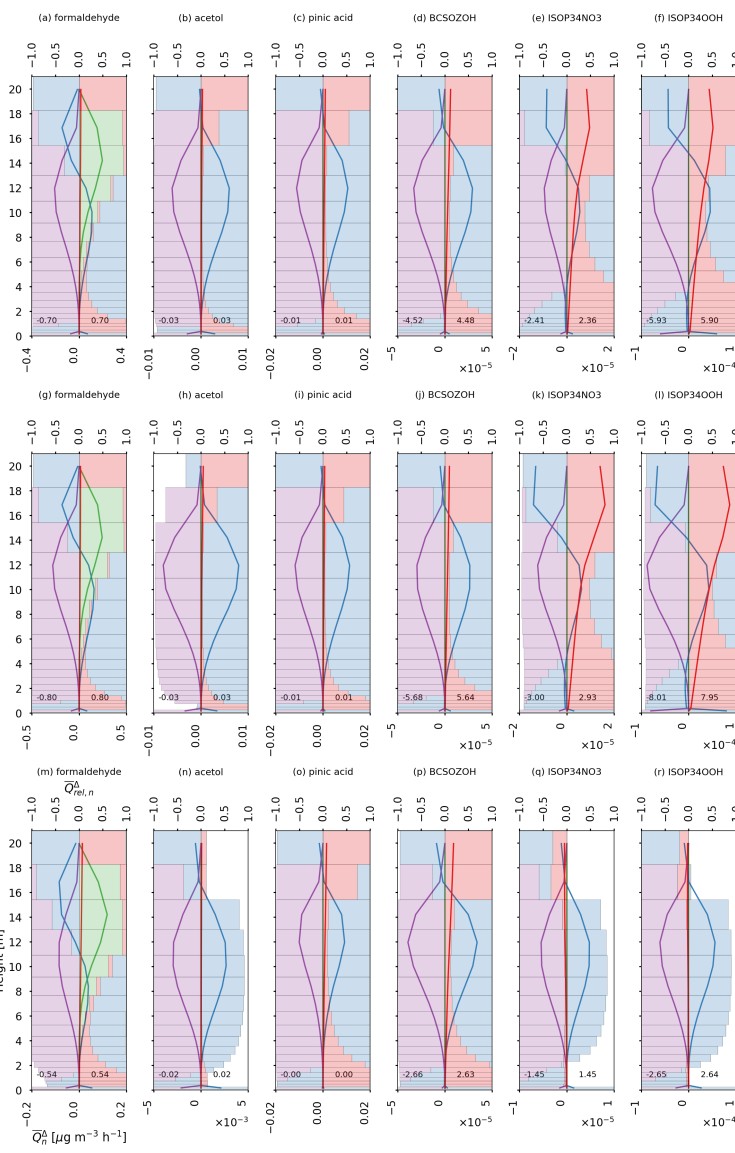

**Figure 9.** The same as Fig. 8 except for formaldehyde, acetol, pinic acid, BCSOZOH, ISOP34NO3 and ISOP34OOH. Note that the bottom numbers for BCSOZOH, ISOP34NO3 and ISOP34OOH also needs to be scaled by $10^{-5}$, $10^{-5}$ and $10^{-4}$, respectively.





## Appendix A: Source versus sink terms

The local change of the trace gas concentration at each model layer is determined by the gas emissions ($Q_{emis}$), chemical production and loss ($Q_{chem}$), gas dry deposition ($Q_{depo}$), and turbulent transport flowing into or out of this layer ($Q_{turb}$). This can be represented by a mass conservation equation:

$$\frac{\partial [X]}{\partial t} = Q^t_{emis} + Q^t_{chem} + Q^t_{depo} + Q^t_{turb} \tag{A1}$$

$$Q^t_{depo} = -[X](\mathrm{LAD} \cdot V_{dveg} + A_s V_{dsoil}) \tag{A2}$$

$$Q^t_{turb} = \frac{\partial}{\partial z}\left( K \frac{\partial [X]}{\partial z} \right) \tag{A3}$$

Here $Q^t_{emis}$ and $Q^t_{chem}$ are directly calculated from the emission module and chemistry module in SOSAA, respectively. The superscript $t$ represents instantaneous quantity. In these calculations, $K$ is the turbulent diffusivity for scalars. The output time step in the model is 30 minutes, so the accumulated values of $Q^t_n$ ($n = emis,\ chem,\ depo,\ turb$) during previous 30 minutes are computed and saved instead of the instantaneous values:

$$\frac{1}{30\mathrm{min}} \int_{t-30\mathrm{min}}^{t} \left( \frac{\partial [X]}{\partial t} = Q^t_{emis} + Q^t_{chem} + Q^t_{depo} + Q^t_{turb} \right) \mathrm{d}t \tag{A4}$$

$$\frac{[X]^t - [X]^{t-30\mathrm{min}}}{30\mathrm{min}} = \overline{Q}^\Delta_{emis} + \overline{Q}^\Delta_{chem} + \overline{Q}^\Delta_{depo} + \overline{Q}^\Delta_{turb} \tag{A5}$$

$$\overline{Q}^\Delta_n = \frac{1}{30\mathrm{min}} \int_{t-30\mathrm{min}}^{t} Q^t_n \mathrm{d}t. \tag{A6}$$

The superscript $\Delta$ represents 30-minute integration period. Moreover, in order to analyze the integrated sources and sinks within the canopy, the in-canopy gas concentration change during previous 30 minutes is calculated as:

$$\frac{1}{h_c} \int_0^{h_c} \left( \frac{[X]^t - [X]^{t-30\mathrm{min}}}{30\mathrm{min}} = \overline{Q}^\Delta_{emis} + \overline{Q}^\Delta_{chem} + \overline{Q}^\Delta_{depo} + \overline{Q}^\Delta_{turb} \right) \mathrm{d}z \tag{A7}$$

$$\overline{Q}^{\Delta,h_c}_n = \frac{1}{hc} \int_0^{hc} \overline{Q}^\Delta_n \mathrm{d}z. \tag{A8}$$

Similarly, the superscripts $\Delta$ and $h_c$ all together represent the integration over previous 30 minutes and from surface to canopy height. Here the positive (negative) $\overline{Q}^{\Delta,h_c}_{turb}$ value indicates the downward (upward) flux at the canopy top resulting in positive (negative) contribution to in-canopy amount of compound $X$.

In addition, the relative contributions of individual sources and sinks are also computed. First, the maximum absolute value between total source and total sink is calculated:

$$Q_{max} = \max(Q_{emis} + \max(Q_{chem}, 0) + \max(Q_{turb}, 0), -(Q_{depo} + \min(Q_{chem}, 0) + \min(Q_{turb}, 0))). \tag{A9}$$




Here we assume that $Q_{emis}$ is always positive while $Q_{depo}$ is always negative. $Q_{chem}$ and $Q_{turb}$ can be either positive or negative. Then the relative contributions are obtained:

$$Q_{rel,n} = \frac{Q_n}{Q_{max}}. \tag{A10}$$

Hence, the values of $Q_{rel,n}$ are in the range of -1 to 1. Here $Q_n$ can be $Q_n^t$, $\overline{Q}_n^{\Delta}$ or $\overline{Q}_n^{\Delta,h_c}$, corresponding to $Q_{rel,n}^t$, $\overline{Q}_{rel,n}^{\Delta}$ or $\overline{Q}_{rel,n}^{\Delta,h_c}$.

*Author contributions.*

*Competing interests.* The authors declare that they have no conflict of interest.

*Acknowledgements.* We acknowledge the support from the Academy of Finland (projects 1118615 and 272041), and the computational resources from CSC – IT Center for Science, Finland. Ditte Taipale acknowledges the support from the European Regional Development Fund (Centre of Excellence EcolChange). Matti P. Rissanen acknowledges the funding from the Academy of Finland (project 299574).





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
