# Peer review of "Boreal forest BVOCs exchange: emissions versus in-canopy sinks"

_Atmospheric Chemistry and Physics, 2017_

## Referee Comment (RC1) · Anonymous Referee #1 · 3 Aug 2017

Zhou et al. have developed a multi-layer gas dry deposition model and present its implementation and application in a one-dimensional chemical transport model in order to quantify source and sink processes of twelve biogenic volatile organic compounds (BVOCs) or BVOC groups in the Hyytiälä forest canopy in July 2010. The explicit description of dry deposition of different gases to vegetation surfaces and soil surfaces is an important extension of the column model SOSAA that helps to disentangle the interactions of biogenic emissions, chemistry, and transport as controlling factors of BVOC concentration changes in and above forest canopies. The study is fully within the scope of Atmospheric Chemistry and Physics and should be published after language-editing and revising the manuscript by taking into account the following comments:

a) One of the main results of the study is the classification of selected BVOCs into

different categories corresponding to the dominant source and sink terms. I recommend to add more discussion about the general validity of the classification of each compound, and to give more guidance about possible changes of this classification in different canopies.

b) I had difficulties following the discussion of many results because of the separation of turbulent transport and gas dry deposition, which is only briefly explained in the appendix. When used to parameterize canopy-scale flux measurements, dry deposition is typically a combination of these processes. Then, the deposition velocity is expressed as the inverse of the quasi-laminar sublayer resistance and the canopy resistance (rb and rc), AND the aerodynamic resistance (ra) derived from the turbulent diffusivity of the scalar of interest. The separation of turbulent transport and gas dry deposition should be clarified in the beginning of section 3.2, and the budget of sources and sinks should be introduced. Also, the definition of a deposition flux by Eq. 1 leads to some ambiguity throughout the manuscript. For example, in Figure 4 on p. 18 fluxes at the canopy top are discussed which are obviously not the deposition flux according to Eq. 1. Also, a downward turbulent flux (e.g. p. 20, line 20) is obviously defined as positive because it is a source with respect to the canopy. However, by micro-meteorological convention a downward turbulent flux (= deposition) would be negative. For clarity, these differences should be clearly pointed out throughout the manuscript.

c) Figures 5-7 seem to indicate that the budget of in-canopy sources and sinks are balanced for almost all compounds for most of the time (when there is no white part in the bar charts). Then, according to Eq. A1 the in-canopy concentration tendency would be zero. Is this true?

d) On page 21, line 6 it is stated that emission sources shift upward during the day, "which implies that PAR and leaf temperature play a comparable role in emission rates besides the LAD". Why does the shift imply a comparable role?

e) In my opinion, there is no additional benefit from the summary in Chapter 5. I

recommend to revise this chapter, removing the purely summarizing parts and focusing on conclusions and synthesis.

Additional minor and technical comments:

p.1, line 15: "Most of the simulated sources and sinks were located...": Without knowing the forest geometry, e.g. the canopy height, absolute heights are not very helpful. I recommend using the height above ground level relative to the canopy height, height/hc (similar to the presentation of Figure 2a and b) throughout the manuscript, and in particular also in Figures 8 and 9 and their discussions.

p.1, line 21: "in-canopy" instead of "in-caonpy"

p.1, lines 22/23: I recommend to divide the last sentence of page 1 into two sentences.

p.2, line 16: A comma is missing after methanol.

p.2, line 19: Revise "... or partly transported into higher atmosphere".

p.2, line 24: "leaf-scale" instead of "leaf-sale"

p.2, line 29: You may also add a reference to Bamberger et al. (2011) Deposition fluxes of terpenes over grassland. J. Geophys. Res. 116, D14305, doi:10.1029/2010JD015457.

p.3, line 10: Could you please add a reference for the GECKO-A model?

p.3, lines 22/23: What do you mean by: "Both of them motivated this study"?

p.3, line 29: "complementary" instead of "complementory"

p.4. line 29: I cannot follow the explanation of the data filtering: A one month-data set of 45-minute averages would consist of almost 1000 data points for each time series. Why did you filter out one data point "from 164 measurement data points"?

p.6, Table 1: In the first and second columns, "delta3-pinene" should read "delta3-carene"

p.11, line 31: Remove "." after "BVOCs"

p.12, Table 3: In the first column, "delta3-pinene" should read "delta3-carene"

p.13, Table 4: In the first column, "delta3-pinene" should read "delta3-carene"

p.14, line 10: The statement that in "the sub-canopy (0.6 m) the incoming PAR was only about 1/4 of that at the canopy top" is not evident from Figure 1b. Is this statement with respect to maximum values or daily sums or something else? Just by looking at Figure 1b, most subcanopy values seem to be around 1/3.

p.14, line 9: Is the accuracy of the precipitation measurement good enough to give a value of 34.64 mm for the accumulated precipitation?

p.15, line 8: "due to the buoyancy" instead of "due to buoancy"

p.15, line 11: "occurrence" instead of "occuring"

p.15, line 16: Is this reference to Section 4.4 pointing to line 26 on page 20? I couldn't find a true demonstration of the potentially important role of leaf wetness in Section 4.4. Also, it should be acknowledged that leaf wetness may play a role at rH < 70 %, e.g. depending on the deliquescent behavior of salts deposited on the vegetation surfaces.

p.16, line 8: Remove "from the ecosystem".

p.17, line 5: I would assume that the standard deviation of the measured data shown in Figure 4b is not only due to measurement uncertainties but also due to day-to-day variability over the course of the month. If this is the case, please rephrase "measurement uncertainties".

p.19, line 4: Revise "still keep similar".

p.19, line 17: "In the second..." instead of "While in the second..."

p.21, line 9: "... inside the canopy compared to the vertical..." instead of "... inside the canopy compared to the effect of vertical..."

p.21, line 21: "most" instead of "a majority portion of"

p.22, line 1: "chemical sources" instead of "chemical productions"

p.22, line 4: "phenomenon" instead of "phenomena"

p.23, lines 5/6: Revise "...need of the application of canopy exchange modelling system..."

p.23, line 15: "causing a slight upward flux" instead of "causing slightly upward flux"

p.24, lines 2-5: This sentence basically repeats p.23, lines 5-7.

---

## Referee Comment (RC2) · Anonymous Referee #2 · 5 Aug 2017

Summary ———-

Zhou et al. present a new column modeling framework for studying BVOC exchange over forest canopies and apply it to investigate the role of emission, deposition, chemistry and turbulent transport over the SMEAR-II boreal site in Finland. The topic is relevant for ACP (though it would also be suitable for GMD) and will be of interest to the community. The model development is a valuable new tool for better understanding land-atmosphere BVOC exchange. Overall the paper is quite well-written.

A couple aspects of the budget discussions are confusing as they are currently presented and need to be clarified (see below). Other than that, my specific comments are minor and mainly suggestions for clarity. They will be straightforward to address and I recommend publication in ACP.

[Figure]

Specific comments –––––-

Is this code being made publicly available to the scientific community? I see no mention of this.

18L13 and Fig 5a, it seems from this that only 20% of emitted methanol makes it out of the canopy. This seems very low. Given that methanol is one of the largest BVOC fluxes to the atmosphere globally, your finding would seem to imply a much larger gross emission flux than is broadly recognized, with the vast majority never making it out of the canopy. It would be worth discussing this in more detail and comparing with other analyses/measurements that have addressed this point.

Fig. 8, I don't see how to reconcile the data shown in Fig 8 with that shown in Fig 5. As an example, in Fig 5a we see that deposition is 5x more important than turbulent transport in removing methanol from the canopy airspace. In Fig 8e we would draw the opposite conclusion, that turbulent flux out of the canopy (blue line) is more important than deposition (purple line). Perhaps it's hard to visually integrate the area under the blue versus purple curves in 8e, but certainly the purple line integral is not 5x the blue line, and it appears to be less. The budget discussions need to be clarified so the reader fully understands what is being shown for the various figures.

5L14, some more information about the implementation of MEGAN within the model should be provided, as there have been different MEGAN versions, and there are different options for things such as treatment of the canopy, inclusion or not of a soil moisture effect, etc. Also the paper being cited (Guenther et al. 2006) only includes a description for isoprene whereas many more compounds are being simulated here.

11L25, the wording here is confusing: "the SEP of methanol is estimated to be ∼75 ng/m2/s by considering both emission and deposition processes . . .". I guess what is meant is that the 75 represents the gross emission flux (as derived from the net flux after accounting for deposition) since you are explicitly simulating deposition as a separate process. But from your wording it sounds like 75 represents the net flux,

which wouldn't make sense. Please clarify language.

16L32, 'the observed large range in formaldehyde fluxes' ... it is not clear why you would say this, as the range in Fig 4f only covers a total of 0.01 ug/m2/h, compared to the other panels which all cover much larger ranges. Do you instead mean the regular occurrence of both positive and negative fluxes?

17L10, the model "does not capture the observed abrupt increase in this downward flux between 12:00LT and 16:00LT". The abruptness of this observed change is suspicious, is it a regular feature or is it the result of extreme data from one particular anomalous day that is showing up in the mean? (I agree however with the text on lines 10-13 that given the small fluxes and large uncertainty bars this discrepancy is within error)

Fig 5, is there no chemical production of acetone from monoterpenes / sesquiterpenes? Terpene oxidation are thought to be a notable source of acetone. Is it just that the timescale for this is long compared to canopy exchange?

Section 5, except for the last paragraph the summary section is just repeating the findings from earlier. This is not that useful. I suggest streamlining this part to just the most important findings and putting more weight on interpretation / synthesis / next steps.

Minor / technical comments ———-

4L28, state why this measurement point was removed

7L1-5, and in the descriptions that follow, it would be helpful to give the units for the different parameters as they are defined.

Table 1, it seems that for OMT and OSQ that 'other minor monoterpenes' and 'other minor sesquiterpenes' should be placed under 'remarks' and C10H16, C15H24 should be placed under formula.

Fig. 2, consider putting a 2nd x-axis on the wind plot for the day and night values to

avoid confusion as the night values are shifted by 2m/s

15L1-8 and Fig. 2, please expand the temperature plot horizontally so the vertical gradients are more apparent. Right now it is too compressed to really see the changes that are discussed on page 15.

15L2-3, since you're using temperature as a proxy for potential temperature I suggest parenthetically pointing out the largest difference between the two (I think ∼0.3K for 36m) and that the observed gradients are larger than this, justifying the approximation

15L8, typo 'buoyancy'

Fig 3, the points for daytime measured SH fluxes are hard to see as they fall under the LH points. Consider changing color or symbol to make them more visible.

Fig. 4, top row missing y-axis title / label

---

## Author Comment (AC1) · 21 Oct 2017

**Reply to comments on "Boreal forest BVOCs exchange: emissions versus in-canopy sinks" by Anonymous Referee # 1**

October 21, 2017

We really appreciate the reviewer for the detailed and valuable comments which help us to improve the manuscript and extend the discussions in it. All the comments are replied below.

1. **Comment: a) One of the main results of the study is the classification of selected BVOCs into different categories corresponding to the dominant source and sink terms. I recommend to add more discussion about the general validity of the classification of each compound, and to give more guidance about possible changes of this classification in different canopies.**

   Reply: Section 4.3 was split into two sections: "4.3 Overview of in-canopy sources and sinks" (from "The simulated monthly averaged ..." to "... given the low OH concentrations (Fig. 5c).") and "4.4 Classification of BVOCs" (from "Therefore the selected BVOCs ..." to "... the one proposed by Wesely (1989)."). More discussions about the general validity of the classification were added in the new section 4.4.

   "The classifications of the featured BVOCs here can also be extended to other canopy types in summertime nearly without any modifications. For example, for isoprene+MBO, monoterpenes and sesquiterpenes the emission is always the only dominant local source within a canopy, although the emission potentials of these BVOCs can vary two or more orders of magnitude between different plant types (Guenther et al., 2012). Therefore, the current classifications for isoprene+MBO, monoterpenes and sesquiterpenes also apply to other canopy types.

   Besides emission and dry deposition, acetaldehyde, methanol, acetone and formaldehyde can be chemically produced from the oxidation of other BVOCs and destroyed via OH oxidation or photolysis (Millet et al., 2010; Jacob et al., 2005; Khan et al., 2015; DiGangi et al., 2011). The chemical production and removal cancel out each other which can finally result in negligible net chemical effect as shown in this study (Fig. 5a). Therefore, the classifications of these four compounds also apply to other canopy types. However, further investigation with numerical simulations are still needed to verify the relative contributions of net chemical effects for different canopy types.

   Chemical production is the only source in the planetary boundary layer for the other non-emitted gases, including acetol, pinic acid, BCSOZOH, ISOP34OOH, ISOP34NO3. They are either produced by direct chemical reactions inside the canopy or transported from above the canopy in all canopy types. Therefore, the classifications of them apply in a general way. "

2. **b) I had difficulties following the discussion of many results because of the separation of turbulent transport and gas dry deposition, which is only briefly explained in the appendix. When used to parameterize canopy-scale flux measurements, dry deposition is typically a combination of these processes. Then, the deposition velocity is expressed as the inverse of the quasi-laminar sublayer resistance and the canopy resistance (rb and rc), AND the aerodynamic resistance (ra) derived**

from the turbulent diffusivity of the scalar of interest. The separation of turbulent transport and gas dry deposition should be clarified in the beginning of section 3.2, and the budget of sources and sinks should be introduced. Also, the definition of a deposition flux by Eq. 1 leads to some ambiguity throughout the manuscript. For example, in Figure 4 on p. 18 fluxes at the canopy top are discussed which are obviously not the deposition flux according to Eq. 1. Also, a downward turbulent flux (e.g. p. 20, line 20) is obviously defined as positive because it is a source with respect to the canopy. However, by micro-meteorological convention a downward turbulent flux (= deposition) would be negative. For clarity, these differences should be clearly pointed out throughout the manuscript.

Reply: 1. SOSAA has a multi-layer canopy module which is different from the big-leaf approach. The turbulent transport of scalars is explicitly computed by the model inside the canopy before they are deposited onto surfaces. Therefore, the aerodynamic resistance ($r_a$) is not explicitly included in our deposition scheme.

2. The introduction of source and sink terms was moved from appendix to the beginning of section 3.2.

3. The fluxes discussed in Fig. 4 are actually the upward turbulent fluxes at the canopy top, which are not the deposition fluxes. Now Eq. 1 was moved to the mass conservation equation and the definition of deposition flux was removed for clarity.

4. The concept used in this study is meant for a consistent discussion of sources and sinks of gases within the canopy, since the canopy is considered as a container.

P5, L25 – P7, L2:

" The gas dry deposition model is based on the $O_3$ dry deposition model described in Zhou et al. (2017). For each model layer, the deposition flux ($F$) of gas $X$ is calculated as

$$F = -[X](\text{LAD}\Delta z V_{dveg} + A_s \Delta z V_{dsoil})$$
$$V_{dveg} = V_{dveg}(r_b, r_{stm}, r_{mes}, r_{cut}, r_{ws}, f_{wet})$$
$$V_{dsoil} = V_{dsoil}(r_{bs}, r_{soil})$$

where $[X]$ is the concentration of gas species $X$, $\Delta z$ is the layer thickness. LAD is the all-sided leaf area density at layer $i$. $A_s$ represents the soil area index (Eq. 17 in Zhou et al., 2017). $V_{dveg}$ is the vegetation layer-specific conductance which is a function of $r_b$ (quasi-laminar boundary layer resistance), $r_{stm}$ (stomatal resistance), $r_{mes}$ (mesophyllic resistance), $r_{cut}$ (dry cuticular resistance), $r_{ws}$ (resistance to leaf wet skin) and $f_{wet}$ (fraction of wet skin on leaf surface) (see Eqs. 8, 10 – 13 in Zhou et al. (2017)). $V_{dsoil}$ is the soil conductance which is a function of $r_{bs}$ (soil boundary layer resistance) and $r_{soil}$ (soil resistance) (see Eq. 9 in Zhou et al. (2017)). "

$\rightarrow$

"The local change of the trace gas concentration at each model layer is determined by the gas emission ($Q_{emis}$), chemical production and loss ($Q_{chem}$), gas dry deposition ($Q_{depo}$), and turbulent transport flowing into or out of this layer ($Q_{turb}$). Here it should be noted that the positive (negative) $Q_{turb}$ is a gas source (sink) term which indicates that the net effect of transportation increases (decreases) the gas concentration within the local layer. All of these processes are included in a mass conservation equation and are computed independently in the

model:

$$\frac{\partial [X]}{\partial t} = Q^t_{emis} + Q^t_{chem} + Q^t_{depo} + Q^t_{turb}$$

$$Q^t_{depo} = -[X](\text{LAD} \cdot V_{dveg} + A_s V_{dsoil})$$

$$Q^t_{turb} = \frac{\partial}{\partial z}\left(K \frac{\partial [X]}{\partial z}\right)$$

$$V_{dveg} = V_{dveg}(r_b, r_{stm}, r_{mes}, r_{cut}, r_{ws}, f_{wet})$$

$$V_{dsoil} = V_{dsoil}(r_{bs}, r_{soil})$$

Here $Q^t_{emis}$ and $Q^t_{chem}$ are directly calculated from the emission module and chemistry module in SOSAA, respectively. The superscript $t$ represents instantaneous quantity. $[X]$ (ng m$^{-3}$) is the concentration of gas species $X$. LAD (m$^2$ m$^{-3}$) is the all-sided leaf area density. $A_s$ (m$^2$ m$^{-3}$) represents the soil area index (Eq. 17 in Zhou et al., 2017). $K$ (m$^2$ s$^{-1}$) is the turbulent diffusivity for scalars. $V_{dveg}$ (m s$^{-1}$) is the vegetation layer-specific conductance which is a function of $r_b$ (quasi-laminar boundary layer resistance; s m$^{-1}$), $r_{stm}$ (stomatal resistance; s m$^{-1}$), $r_{mes}$ (mesophyllic resistance; s m$^{-1}$), $r_{cut}$ (dry cuticular resistance; s m$^{-1}$), $r_{ws}$ (resistance to leaf wet skin; s m$^{-1}$) and $f_{wet}$ (fraction of wet skin on leaf surface; dimensionless) (see Eqs. 8, 10 – 13 in Zhou et al., 2017). $V_{dsoil}$ is the soil conductance which is a function of $r_{bs}$ (soil boundary layer resistance; s m$^{-1}$) and $r_{soil}$ (soil resistance; s m$^{-1}$) (see Eq. 9 in Zhou et al., 2017). "

P29, L1: "Appendix A: Source versus sink terms" → "Appendix A: Accumulated and integrated source and sink terms".

P29, L2–9: "The local change ... $K$ is the turbulent diffusivity for scalars." was removed.

3. **Comment: c) Figures 5-7 seem to indicate that the budget of in-canopy sources and sinks are balanced for almost all compounds for most of the time (when there is no white part in the bar charts). Then, according to Eq. A1 the in-canopy concentration tendency would be zero. Is this true?**

Reply: Yes, it is true. All the source and sink terms are accumulated with time, e.g., emission and dry deposition. Therefore, the net in-canopy concentration tendency is neglected compared to these sources and sinks. For example, the in-canopy concentration change of methanol during the whole month is 2114.6 $\mu$g m$^{-3}$ due to accumulated emissions, -3.5 $\mu$g m$^{-3}$ due to chemical reactions, -406.5 $\mu$g m$^{-3}$ due to turbulent transport, -1701.8 $\mu$g m$^{-3}$ due to dry deposition. And the final concentration change is about 2.8 $\mu$g m$^{-3}$ which is only 0.1% of the emission source.

4. **Comment: d) On page 21, line 6 it is stated that emission sources shift upward during the day, "which implies that PAR and leaf temperature play a comparable role in emission rates besides the LAD". Why does the shift imply a comparable role?**

Reply: Besides LAD, gas emissions are mainly dependent on leaf temperature and/or PAR, which are attenuated inside the canopy and thus decreasing gas emissions. If they play a minor role compared to LAD, the profiles of emission sources would be similar with the LAD profile. However, the peak heights of emission sources are about 50% higher than the LAD peak height, which indicates that PAR and leaf temperature also play an important role.

5. **Comment: e) In my opinion, there is no additional benefit from the summary in Chapter 5. I recommend to revise this chapter, removing the purely summarizing parts and focusing on conclusions and synthesis.**

Reply: The summary part was rewritten.

[revised manuscript text omitted]

6. **Comment: p.1, line 15: ”Most of the simulated sources and sinks were located...”: Without knowing the forest geometry, e.g. the canopy height, absolute heights are not very helpful. I recommend using the height above ground level relative to the canopy height, height/hc (similar to the presentation of Figure 2a and b) throughout the manuscript, and in particular also in Figures 8 and 9 and their discussions.**

Reply:

P1, L15: "about 4 m" $\rightarrow$ "about 0.2 $h_c$ (canopy height)".

P1, L15: "about 8 m" $\rightarrow$ "about 0.4 $h_c$".

P1, L17: "about 14 - 16 m" $\rightarrow$ "about 0.8 - 0.9 $h_c$".

P1, L17: "than 10 m" $\rightarrow$ "than 0.6 $h_c$".

P21, L22: "about 8 m" $\rightarrow$ "about 0.4 $h_c$".

P21, L23: "about 4 m" $\rightarrow$ "about 0.2 $h_c$".

P21, L24: "below 8 or 4 m" $\rightarrow$ "below 0.4 or 0.2 $h_c$".

P23, L19-23: The whole summary section was rewritten, so the height levels were not converted to canopy height here.

Figures 8 and 9: The y axes were changed to "Height/$h_c$". Several x scales were modified to zoom in the plots in x direction (figures are placed at last).

7. **Comment: p.1, line 21: ”in-canopy” instead of ”in-caonpy”**

Reply: P1, L21: "in-caonpy" $\rightarrow$ "in-canopy".

8. **Comment: p.1, lines 22/23: I recommend to divide the last sentence of page 1 into two sentences.**

Reply: P1, L22–23: "Twelve featured BVOCs or BVOC groups were analyzed in this study, more compounds could also be investigated similarly by being classified into the five categories."

$\rightarrow$

"Twelve featured BVOCs or BVOC groups were analyzed in this study. Other compounds could also be investigated similarly by being classified into these five categories.".

9. **Comment: p.2, line 16: A comma is missing after methanol.**

Reply: P2, L16: "methanol acetaldehyde" $\rightarrow$ "methanol, acetaldehyde".

10. **Comment: p.2, line 19: Revise ”... or partly transported into higher atmosphere”.**

Reply: P2, L19: ", or partly transported into higher atmosphere." $\rightarrow$ ", or transported throughout the planetary boundary layer".

11. **Comment: p.2, line 24: ”leaf-scale” instead of ”leaf-sale”**

Reply: P2, L24: "leaf-sale" $\rightarrow$ "leaf-scale".

12. **Comment: p.2, line 29: You may also add a reference to Bamberger et al. (2011) Deposition fluxes of terpenes over grassland. J. Geophys. Res. 116, D14305, doi:10.1029/2010JD015457.**

Reply: P2, L30: "... in field measurements."

$\rightarrow$

"... in field measurements. Bamberger et al. (2011) observed the deposition fluxes of monoterpenes, sesquiterpenes and oxygenated terpenes over a temperate mountain grassland in an alpine valley after a hailstorm."

13. **Comment: p.3, line 10: Could you please add a reference for the GECKO-A model?**

Replay: A reference was added and a typo was also corrected in the same sentence.

P3, L10–12: "The models GECKO-A (Generator of Explicit Chemistry and Kinetics of Organics in the Atmosphere) and GROMHE (Raventos-Duran et al., 2010, GROup contribution Method for Henry's law Estimate;) ..."

$\rightarrow$

"The models GECKO-A (Generator of Explicit Chemistry and Kinetics of Organics in the Atmosphere; Aumont et al., 2005) and GROMHE (GROup contribution Method for Henry's law Estimate; Raventos-Duran et al., 2010) ..."

14. **Comment: p.3, lines 22/23: What do you mean by: "Both of them motivated this study"?**

Reply: As we mentioned in the same paragraph: the models, which calculated the deposition sinks of a large number of BVOCs, were big-leaf models. The other models, which were multi-layer canopy models, only considered a small amount of BVOCs (usually less than 100 gas species). So these gaps motivated this study. This sentence here was not clear, so we deleted it and modified the first sentence in next paragraph.

P3, L22-23: "Moreover, detailed deposition contributions for BVOCs have not been analysed. Both of them motivated this study."

$\rightarrow$

"Moreover, detailed deposition contributions for BVOCs have not been analysed."

P3, L24: "A multi-layer gas dry deposition model has been developed in this study based on several models in previous studies ..."

$\rightarrow$

"In order to fill the gaps mentioned above, a multi-layer gas dry deposition model has been developed in this study based on several models in previous studies ..."

15. **Comment: p.3, line 29: "complementary" instead of "complementory"**

Reply: P3, L29: "complementory" $\rightarrow$ "complementary"

16. **Comment: p.4. line 29: I cannot follow the explanation of the data filtering: A one month-data set of 45-minute averages would consist of almost 1000 data points for each time series. Why did you filter out one data point "from 164 measurement data points"?**

Reply: In 2010, BVOC concentrations were measured by PTR-MS at six levels (4.2 m, 8.4 m, 16.8 m, 33.6 m, 50.4 m and 67.2 m) one by one in one single measurement cycle (6 minutes). Then nine cycles including one cycle for determining instrumental background were conducted every third hour. The 45-minute averages of BVOC concentrations were computed from nine cycles. Finally, the BVOC fluxes were calculated from the 45-minute average data with the surface-layer-profile method (Rantala et al., 2014, 2015). So the flux data were computed every third hour, containing at most 248 points in July, 2010. 164 measurement data were available after excluding the periods when instruments worked improperly. This sentence was modified to be more clear.

P4, L29: "The fluxes of BVOCs, based on 45-minute averages, were computed with the surface-layer-profile method ..."

$\rightarrow$

"The fluxes of BVOCs, based on 45-minute averages of BVOC concentrations, were computed every third hour with the surface-layer-profile method ...".

17. **Comment: p.6, Table 1: In the first and second columns, "delta3-pinene" should read "delta3-carene"**

    Reply: P6, Table 1: "$\Delta^3$-pinene" $\rightarrow$ "$\Delta^3$-carene".

18. **Comment: p.11, line 31: Remove "." after "BVOCs"**

    Reply: P11, L31: "BVOCs. (Table ..." $\rightarrow$ "BVOCs (Table ..."

19. **Comment: p.12, Table 3: In the first column, "delta3-pinene" should read "delta3-carene"**

    Reply: P12, Table 3: "$\Delta^3$-pinene" $\rightarrow$ "$\Delta^3$-carene".

20. **Comment: p.13, Table 4: In the first column, "delta3-pinene" should read "delta3-carene"**

    Reply: P13, Table 4: "$\Delta^3$-pinene" $\rightarrow$ "$\Delta^3$-carene".

21. **Comment: p.14, line 10: The statement that in "the sub-canopy (0.6 m) the incoming PAR was only about 1/4 of that at the canopy top" is not evident from Figure 1b. Is this statement with respect to maximum values or daily sums or something else? Just by looking at Figure 1b, most subcanopy values seem to be around 1/3.**

    Reply: It describes the monthly average PAR values. The text was modified.

    P14, L7: "the incoming PAR was only about 1/4" $\rightarrow$ "the monthly-averaged incoming PAR was only 1/4".

22. **Comment: p.14, line 9: Is the accuracy of the precipitation measurement good enough to give a value of 34.64 mm for the accumulated precipitation?**

    Reply: The measured 1-minute accumulated precipitation data from SMEAR II data server (downloaded from https://avaa.tdata.fi/web/smart/smear/search) has the precision of 0.01 mm. However, according to the manual of the instrument Vaisala FD12P (https://www.manualslib.com/manual/538824/Vaisala-Fd12p.html), the accuracy of precipitation intensity is $\pm 30\%$ in the range of 0.5 to 20 mm h$^{-1}$. So the accuracy of accumulated precipitation during previous 1 minute is in the range of 0.008 to 0.3 mm. We modified the number with less precision.

    P14, L9: "34.64 mm" $\rightarrow$ "35 mm".

23. **Comment: p.15, line 8: "due to the buoyancy" instead of "due to buoancy"**

    Reply: P15, L8: "due to buoancy" $\rightarrow$ "due to the buoyancy".

24. **Comment: p.15, line 11: "occurrence" instead of "occuring"**

    Reply: P15, L11: "occuring" $\rightarrow$ "occurrence".

25. **Comment: p.15, line 16: Is this reference to Section 4.4 pointing to line 26 on page 20? I couldn't find a true demonstration of the potentially important role of leaf wetness in Section 4.4. Also, it should be acknowledged that leaf wetness may play a role at RH < 70 %, e.g. depending on the deliquescent behavior of salts deposited on the vegetation surfaces.**

Reply: We removed the latter part of the sentence and added the impact of deliquescence in the end of the paragraph.

P15, L14-16: "Therefore, the observed RH values inside the canopy were used to parametrise $f_{wet}$ when calculating the deposition velocity to represent a more realistic leaf wetness condition, also since this leaf wetness plays a potentially important role in BVOC exchange as we demonstrate in further details below in Section 4.4."

$\rightarrow$

"Therefore, the observed RH values inside the canopy were used to parametrise $f_{wet}$ when calculating the deposition velocity to represent a more realistic leaf wetness condition. It should be noted here that although RH = 70% is chosen as a threshold of the occurrence of leaf wetness in the model, the leaf wetness may already play a role when RH < 70%, e.g., due to the deliquescent effect of deposited salt on the vegetation surfaces."

26. **Comment: p.16, line 8: Remove "from the ecosystem".**

    Reply: P16, L8: "from the ecosystem into the soil" $\rightarrow$ "into the soil".

27. **Comment: p.17, line 5: I would assume that the standard deviation of the measured data shown in Figure 4b is not only due to measurement uncertainties but also due to day-to-day variability over the course of the month. If this is the case, please rephrase "measurement uncertainties".**

    Reply: P17, L5: "measurement uncertainties" $\rightarrow$ "measurement uncertainties and day-to-day variation".

28. **Comment: p.19, line 4: Revise "still keep similar".**

    Reply: P19, L4: "... emitted compounds still keep similar except for ..." $\rightarrow$ "... emitted compounds are similar with that at daytime except for ...".

29. **Comment: p.19, line 17: "In the second..." instead of "While in the second..."**

    Reply: P19, L17: "While in the second category, ..." $\rightarrow$ "In the second category, ...".

30. **Comment: p.21, line 9: "... inside the canopy compared to the vertical..." instead of "... inside the canopy compared to the effect of vertical..."**

    Reply: P21, L9: "... inside the canopy compared to the effect of vertical temperature gradient ..." $\rightarrow$ "... inside the canopy compared to the vertical temperature gradient ...".

31. **Comment: p.21, line 21: "most" instead of "a majority portion of"**

    Reply: P21, L21: "a majority portion of" $\rightarrow$ "most".

32. **Comment: p.22, line 1: "chemical sources" instead of "chemical productions"**

    Reply: P22, L1: "chemical productions" $\rightarrow$ "chemical sources".

33. **Comment: p.22, line 4: "phenomenon" instead of "phenomena"**

    Reply: "phenomena" $\rightarrow$ "phenomenon".

34. **Comment: p.23, lines 5/6: Revise "...need of the application of canopy exchange modelling system..."**

    Reply: The whole summary section was rewritten (see Comment 5)

35. **Comment: p.23, line 15: "causing a slight upward flux" instead of "causing slightly upward flux"**

    Reply: The whole summary section was rewritten (see Comment 5)

36. **Comment: p.24, lines 2-5: This sentence basically repeats p.23, lines 5-7.**

    Reply: The whole summary section was rewritten (see Comment 5)

**References**

[revised manuscript text omitted]

---

## Author Comment (AC2) · 21 Oct 2017

**Reply to comments on "Boreal forest BVOCs exchange: emissions versus in-canopy sinks" by Anonymous Referee # 2**

October 21, 2017

We really appreciate the reviewer's detailed and valuable comments which help to improve our manuscript and clarify our discussions. All the comments are replied below.

1. **Comment: Is this code being made publicly available to the scientific community? I see no mention of this.**

   Reply: We will attach the python code of calculating the Henry's law constants and reactivity factors which is not a part of SOSAA currently. The whole SOSAA code is available by contacting Michael Boy (michael.boy@helsinki.fi) or Zhou Putian (putian.zhou@helsinki.fi). A two-month visit to the group of Dr. Boy is required to achieve the SOSAA model to ensure that the person is able to understand the basics of the code.

2. **Comment: 18L13 and Fig 5a, it seems from this that only 20% of emitted methanol makes it out of the canopy. This seems very low. Given that methanol is one of the largest BVOC fluxes to the atmosphere globally, your finding would seem to imply a much larger gross emission flux than is broadly recognized, with the vast majority never making it out of the canopy. It would be worth discussing this in more detail and comparing with other analyses/measurements that have addressed this point.**

   Reply: More discussion about methanol budget was added.

   P18, L14: "Hence their fluxes are bidirectional in the simulation (Figs. 4c-f)."

   $\rightarrow$

   "Hence their fluxes are bidirectional in the simulation (Figs. 4c-f). The results indicate that a large portion of methanol molecules are deposited inside the canopy instead of being transported out of the canopy, which were also noticed by other studies. Karl et al. (2005) found methanol was deposited mostly in the lower canopy part during daytime and uptaken significantly inside the canopy at nighttime in a loblolly pine forest in July, 2003. Laffineur et al. (2012) even reported net daily negative methanol fluxes in a temperate mixed forest in summer during 2009 and 2010. At SMEAR II, Rantala et al. (2015) showed that from April to September during 2010 to 2013 the ratio between the cumulative deposition and the cumulative emission was slightly lower than 40%, which is about half of that in this study (80%). This discrepancy may result from the soil deposition explicitly calculated in this study, which is about 42% of the overall dry deposition sink of methanol. "

3. **Comment: Fig. 8, I don't see how to reconcile the data shown in Fig 8 with that shown in Fig 5. As an example, in Fig 5a we see that deposition is 5x more important than turbulent transport in removing methanol from the canopy airspace. In Fig 8e we would draw the opposite conclusion, that turbulent flux out of the canopy (blue line) is more important than deposition (purple line). Perhaps**

**it's hard to visually integrate the area under the blue versus purple curves in 8e, but certainly the purple line integral is not 5x the blue line, and it appears to be less. The budget discussions need to be clarified so the reader fully understands what is being shown for the various figures.**

Reply: In Fig. 8e methanol is transported out of the upper canopy layers. It can be transported either upward out of the canopy or downward into the lower canopy. In the lower part of the canopy, turbulent transport increases the methanol concentration as a source to compensate the deposition (-1.17 $\mu$g m$^{-3}$ h$^{-1}$) onto soil surface. Therefore, turbulent transport acts as a sink term in the upper part of canopy and a source term in the lower part, which counteract each other. Therefore the overall effect of turbulent transport is -0.66 $\mu$g m$^{-3}$ h$^{-1}$ and the value for deposition is -2.77 $\mu$g m$^{-3}$ h$^{-1}$ which is about 4.2 times larger. This is consistent with the results shown in Fig. 5a.

4. **Comment: 5L14, some more information about the implementation of MEGAN within the model should be provided, as there have been different MEGAN versions, and there are different options for things such as treatment of the canopy, inclusion or not of a soil moisture effect, etc. Also the paper being cited (Guenther et al. 2006) only includes a description for isoprene whereas many more compounds are being simulated here.**

Reply: Some texts were added to clarify the MEGAN module used in this study.

P5, L13-15: "The BVOC emissions from the forest ecosystem are computed by MEGAN (Model of Emissions of Gases and Aerosols from Nature; Guenther et al., 2006)."

$\rightarrow$

"The BVOC emissions from the forest ecosystem are computed by a modification version of MEGAN 2.04 (Model of Emissions of Gases and Aerosols from Nature; Guenther et al., 2006) which was described in details in Mogensen et al. (2015) and Zhou et al. (2017)."

P11, L9: "The emissions of 15 organic compounds are included in current simulations, which are $\alpha$-pinene, $\beta$-pinene, $\Delta$3-carene, limonene, 1,8-cineole, OMT, $\beta$-caryophyllene, farnesene, OSQ, isoprene, MBO, methanol, acetaldehyde, acetone, formaldehyde. Their standard emission potentials (SEPs) for July, 2010 at SMEAR II ..."

$\rightarrow$

"The emissions of 15 organic compounds ($\alpha$-pinene, $\beta$-pinene, $\Delta$3-carene, limonene, 1,8-cineole, OMT, $\beta$-caryophyllene, farnesene, OSQ, isoprene, MBO, methanol, acetaldehyde, acetone, formaldehyde) are computed in current MEGAN module according to the canopy structure described in Sec. 2.1. In this study only the emissions from the Scots pine are considered (Mogensen et al., 2015). The soil moisture is large enough during the whole month so that the activity factor for soil moisture is always equal to 1.0. The standard emission potentials (SEPs) of these 15 compounds for July, 2010 at SMEAR II ..." "

5. **Comment: 11L25, the wording here is confusing: "the SEP of methanol is estimated to be $\sim$ 75 ng/m2/s by considering both emission and deposition processes . . .". I guess what is meant is that the 75 represents the gross emission flux (as derived from the net flux after accounting for deposition) since you are explicitly simulating deposition as a separate process. But from your wording it sounds like 75 represents the net flux, which wouldn't make sense. Please clarify language.**

Reply: Yes, here 75 ng m$^{-2}$ s$^{-1}$ represents the emission flux. Rantala et al. (2015) applied an exchange algorithm including emission ($E$) and deposition ($D$) to parametrise the net flux of methanol ($F$):

$$F = E - D$$

Here $E$ is dependent on methanol SEP and other meteorological conditions, while $D$ has nothing to do with SEP. Then the SEP of methanol was calculated from measurement data ($F$ and other meterological quantities) as 75 ng m$^{-2}$ s$^{-1}$ for July at SMEAR II, which was used in our study to calculate the methanol emissions. Text here was modified.

P11, L25-26: "The SEP of methanol is estimated to be $\sim$ 75 ng m$^{-2}$ s$^{-1}$ by considering both emission and deposition processes for July at SMEAR II in Rantala et al. (2015)."

$\rightarrow$

"The SEP of methanol is set to $\sim$ 75 ng m$^{-2}$ s$^{-1}$ as suggested in Rantala et al. (2015)."

6. **Comment: 16L32, 'the observed large range in formaldehyde fluxes' . . . it is not clear why you would say this, as the range in Fig 4f only covers a total of 0.01 ug/m2/h, compared to the other panels which all cover much larger ranges. Do you instead mean the regular occurrence of both positive and negative fluxes?**

Reply: Yes, we meant bi-directional fluxes. The text was modified to make it more clear.

P16, L32: "The observed large range in formaldehyde fluxes also indicate ..."

$\rightarrow$

"The observed apparent bi-directional formaldehyde fluxes also indicate ..."

7. **Comment: 17L10, the model "does not capture the observed abrupt increase in this downward flux between 12:00LT and 16:00LT". The abruptness of this observed change is suspicious, is it a regular feature or is it the result of extreme data from one particular anomalous day that is showing up in the mean? (I agree however with the text on lines 10-13 that given the small fluxes and large uncertainty bars this discrepancy is within error)**

Reply: The elevated downward flux smaller than -0.005 $\mu$g m$^{-2}$ s$^{-1}$ (absolute value larger than 0.005 $\mu$g m$^{-2}$ s$^{-1}$) in the afternoon occurred in 10 days during the whole month. Therefore, it is a regular feature. However, the accuracy on PTR-MS measurements of formaldehyde is questionable because the proton affinity of water and formaldehyde is almost the same as we have mentioned in section 2.2.2. Therefore, we can not say if this feature is natural or results from measurement uncertainties. More measurement data are needed to further clarify the diurnal variation of formaldehyde flux.

8. **Comment: Fig 5, is there no chemical production of acetone from monoterpenes / sesquiterpenes? Terpene oxidation are thought to be a notable source of acetone. Is it just that the timescale for this is long compared to canopy exchange?**

Reply: Inside the canopy, about 86% of the emitted monoterpenes are transported out of the canopy and only 9% of them are oxidized. Moreover, although 70% of the emitted sesquiterpenes are oxidized, the emission rate of sesquiterpenes is only 20% of monoterpenes. Therefore, the oxidation products from these two groups of precursor gases do not provide enough chemical production of acetone inside the canopy compared to emissions.

9. **Comment: Section 5, except for the last paragraph the summary section is just repeating the findings from earlier. This is not that useful. I suggest streamlining this part to just the most important findings and putting more weight on interpretation / synthesis / next steps.**

Reply: The summary part was rewritten.

[revised manuscript text omitted]

10. **Comment: 4L28, state why this measurement point was removed**

Reply: P4, L28-29: "Finally, for each compound one data point was filtered out from 164 measurement data points."

$\rightarrow$

"Finally, for each compound one data point was filtered out from 164 measurement data points due to $\zeta > 1$."

11. **Comment: 7L1-5, and in the descriptions that follow, it would be helpful to give the units for the different parameters as they are defined.**

Reply: The units in the whole section 3.2.1 were added. The first paragraph was rewritten according to the comments from Referee #1.

"The local change of the trace gas concentration at each model layer is determined by the gas emission ($Q_{emis}$), chemical production and loss ($Q_{chem}$), gas dry deposition ($Q_{depo}$), and turbulent transport flowing into or out of this layer ($Q_{turb}$). Here it should be noted that the positive (negative) $Q_{turb}$ is a gas source (sink) term which indicates that the net effect of transportation increases (decreases) the gas concentration within the local layer. All of these processes are included in a mass conservation equation and are computed independently in the model:

$$\frac{\partial [X]}{\partial t} = Q_{emis}^t + Q_{chem}^t + Q_{depo}^t + Q_{turb}^t$$

$$Q_{depo}^t = -[X](\text{LAD} \cdot V_{dveg} + A_s V_{dsoil})$$

$$Q_{turb}^t = \frac{\partial}{\partial z}\left(K \frac{\partial [X]}{\partial z}\right)$$

$$V_{dveg} = V_{dveg}(r_b, r_{stm}, r_{mes}, r_{cut}, r_{ws}, f_{wet})$$

$$V_{dsoil} = V_{dsoil}(r_{bs}, r_{soil})$$

Here $Q_{emis}^t$ and $Q_{chem}^t$ are directly calculated from the emission module and chemistry module in SOSAA, respectively. The superscript $t$ represents instantaneous quantity. $[X]$ (ng m$^{-3}$) is the concentration of gas species $X$. LAD (m$^2$ m$^{-3}$) is the all-sided leaf area density. $A_s$ (m$^2$ m$^{-3}$) represents the soil area index (Eq. 17 in Zhou et al., 2017). $K$ (m$^2$ s$^{-1}$) is the turbulent diffusivity for scalars. $V_{dveg}$ (m s$^{-1}$) is the vegetation layer-specific conductance which is a function of $r_b$ (quasi-laminar boundary layer resistance; s m$^{-1}$), $r_{stm}$ (stomatal resistance; s m$^{-1}$), $r_{mes}$ (mesophyllic resistance; s m$^{-1}$), $r_{cut}$ (dry cuticular resistance; s m$^{-1}$), $r_{ws}$ (resistance to leaf wet skin; s m$^{-1}$) and $f_{wet}$ (fraction of wet skin on leaf surface; dimensionless) (see Eqs. 8, 10 – 13 in Zhou et al. (2017)). $V_{dsoil}$ is the soil conductance which is a function of $r_{bs}$ (soil boundary layer resistance; s m$^{-1}$) and $r_{soil}$ (soil resistance; s m$^{-1}$) (see Eq. 9 in Zhou et al. (2017)). "

P7, L12: "S$_c$" $\rightarrow$ "S$_c$ (dimensionless)".

P7, L13: "molecular diffusivity ($D_X$)" $\rightarrow$ "molecular diffusivity ($D_X$; m$^2$ s$^{-1}$)".

P7, L14: "the molar mass ratio" $\rightarrow$ "the molar mass (g mol$^{-1}$)".

P7, L15: "U" $\rightarrow$ "U (m s$^{-1}$)".

P8, L1: "$\delta_0$" $\rightarrow$ "$\delta_0$ (m)".

P8, L2: "$u_{*g}$" $\rightarrow$ "$u_{*g}$ (m s$^{-1}$)".

P8, L15: "$f_0$" $\rightarrow$ "$f_0$ (dimensionless)".

P8, L16: "$T_l$" $\rightarrow$ "$T_l$ (K)".

12. **Comment: Table 1, it seems that for OMT and OSQ that 'other minor monoterpenes' and 'other minor sesquiterpenes' should be placed under 'remarks' and C10H16, C15H24 should be placed under formula.**

Reply: Table 1: "other minor monoterpenes" and "other minor sesquiterpenes" were moved to the Remark column. "$C_{10}H_{16}$" and "$C_{15}H_{24}$" were added in the Formula column.

13. **Comment: Fig. 2, consider putting a 2nd x-axis on the wind plot for the day and night values to avoid confusion as the night values are shifted by 2m/s**

    Reply: Figure 2a was replotted with a 2nd x-axis for the nighttime wind values (figures are placed at last).

14. **Comment: 15L1-8 and Fig. 2, please expand the temperature plot horizontally so the vertical gradients are more apparent. Right now it is too compressed to really see the changes that are discussed on page 15.**

    Reply: Figure 2b was replotted with larger extension in x direction (figures are placed at last).

15. **Comment: 15L2-3, since you're using temperature as a proxy for potential temperature I suggest parenthetically pointing out the largest difference between the two (I think $\sim 0.3K$ for 36m) and that the observed gradients are larger than this, justifying the approximation**

    Reply: P15, L2-3: "Hence, the air temperature can be assumed to be the potential temperature within this vertical range."

    $\rightarrow$

    "Hence, the air temperature can be assumed to be the potential temperature within this vertical range (the largest difference between potential temperature and $T$ at 36 m is about 0.35 K which is smaller than the observed gradients)."

16. **Comment: 15L8, typo 'buoyancy'**

    Reply: P15, L8: "buoancy" $\rightarrow$ "buoyancy".

17. **Comment: Fig 3, the points for daytime measured SH fluxes are hard to see as they fall under the LH points. Consider changing color or symbol to make them more visible.**

    Reply: Figure 3 was replotted to make SH more clear (figures are placed at last).

18. **Comment: Fig. 4, top row missing y-axis title / label**

    Reply: A description was added in captions for Figs. 4, 6 and 8: "The x labels and y labels of the left bottom subfigure also apply to all the other subfigures."

[Figure]

Figure 2: Modeled (green solid line for daytime, green dashed line for nighttime) and measured (yellow solid circle for daytime, yellow empty circle for nighttime) profiles of (a) horizontal wind speed (windh) and (b) air temperature ($T$). The ranges of $\pm 1$ SD (standard deviation) of modeled and measured data are marked as shades and error bars. The height is normalised by canopy height ($h_c$). The monthly-averaged diurnal cycles of modeled (green line) and measured (yellow dots) (c) friction velocity ($u_*$) at 23 m and (d) mean RH inside the canopoy are also plotted. The ranges of $\pm 1$ SD of modeled and measured data are marked as shades and vertical lines.

[Figure]

Figure 3: The modeled (solid lines) and measured (points) monthly-averaged diurnal cycles of sensible heat flux ($SH$, blue), latent heat flux ($LE$, green), soil heat flux ($G_{soil}$, yellow) and upward net radiation ($R_{net}$, purple, the observed $R_{net}$ is at 67 m). The ranges of $\pm 1$ standard deviation for modeled and measured data are marked by shaded areas and vertical lines, respectively.